# New Agents in the Treatment of Psychiatric Disorders: What Innovations and in What Areas of Psychopathology?

**DOI:** 10.3390/ph18050665

**Published:** 2025-04-30

**Authors:** Paola Bozzatello, Roberta Novelli, Rebecca Schisano, Claudio Brasso, Paola Rocca, Silvio Bellino

**Affiliations:** Department of Neuroscience, University of Turin, Via Cherasco 15, 10126 Turin, Italy; roberta.novelli@unito.it (R.N.); rebecca.schisano@unito.it (R.S.); claudio.brasso@unito.it (C.B.); paola.rocca@unito.it (P.R.); silvio.bellino@unito.it (S.B.)

**Keywords:** new agents, schizophrenia, bipolar disorder, mood disorders, major depression, obsessive-compulsive disorder, anxiety disorders, tolerability, efficacy

## Abstract

Psychiatric disorders are a major cause of illness in the world. Unfortunately, many patients are resistant to treatment and present serious complications. Schizophrenia is refractory to treatment in about one-third of patients. Antidepressants are effective in about half of patients. Suicidal ideation is an increasing issue in patients with mixed features in bipolar disorder (BD). Therefore, there is a need to develop and test new drugs or new indications of available medications for the treatment of psychiatric disorders through evidence-based investigations. This narrative review aims to present the molecules approved by the main drug agencies, the Food and Drug Administration (FDA) and the European Medicines Agency (EMA), from 2018 to date, along with new indications and new formulations of existing medications. We searched PubMed for new drugs approved for schizophrenia, BD, major depressive disorder (MDD), anxiety disorders, and obsessive-compulsive disorder (OCD). We evaluated their clinical benefits, safety, and tolerability profiles. Finally, we considered studies on the main molecules that have shown initial evidence of efficacy and are in the process of obtaining approval. Our search suggested that a new antipsychotic, lumateperone, and two drug combinations, olanzapine/samidorphan (OLZ/SAM) and xanomeline/trospium (KarXT), were approved for schizophrenia. In addition, some new methods of administration—monthly risperidone administration, subcutaneous risperidone administration, and transdermal asenapine administration—obtained approval from the main drug agencies. Lumateperone and OLZ/SAM were also approved in BD. Esketamine, a compound that modulates glutamatergic transmission, was approved to treat treatment-resistant depression and acute suicidal ideation. The dextromethorphan/bupropion combination was approved for MDD. Two new agents, brexanolone and zuranolone, were approved for treatment of postpartum depression. On the other hand, no new drugs received approval for anxiety disorders or OCD. In summary, some new psychotropic medications have been developed, in particular with the aim to improve the symptoms of resistant patients and to decrease the incidence of adverse effects. It is necessary to continue testing the effectiveness of new compounds in methodologically rigorous studies.

## 1. Introduction

Mental disorders are ranked among the top ten causes of disease globally, and a significant proportion of patients demonstrate high levels of treatment resistance [1,2]. Of all mental disorders, major depressive disorder (MDD) and anxiety disorders are the most common causes of the global disease burden. Schizophrenia and bipolar disorder (BD) are less common in the world population, but the disability associated with psychosis costs 10 times more than general patient care [1,3]. Although pharmacotherapy offers symptom improvements in many patients, antidepressants are effective in only about half of treated patients, and schizophrenia is refractory to treatment in about one-third of patients [3,4]. In 2019, across all age groups, depressive disorders were the thirteenth most frequent cause of disability-adjusted life years (DALYs) and anxiety disorders the twenty-fourth. DALYs are a unit of measurement that quantifies the impact of a disease on a person’s health. The percentage change in DALYs for depressive and anxiety disorders increased from 1990 to 2019 by 61.1% and 53.7%, respectively. Furthermore, in the 10–24 age group, depressive disorders are the fourth cause of DALYs, soon after “self-harm”. Despite the higher number of DALYs in the last years, the development of new pharmacological agents in psychiatry proceeds at a slower pace than in other areas of medicine with a similar impact in terms of disability (i.e., cancer) [1,5]. In order to overcome this disparity, we have to promote the development of new drugs and further clinical investigations to test the efficacy of these compounds in particular populations (i.e., treatment-resistant patients, young and/or old samples, psychiatric disturbances postpartum). A substantial proportion of the compounds recently investigated for the treatment of mental disorders originate from drug repurposing strategies involving established pharmacological agents now being evaluated for novel therapeutic indications, including the formalization of previously off-label applications [2,5].

The present narrative review is aimed to provide an updated overview of the main advances in pharmacological treatments of psychiatric disorders. We searched for and reviewed, both in terms of efficacy and safety, new psychiatric drugs approved by the major drug agencies from 2018 to 2025 for schizophrenia, BD, MDD, anxiety disorders, and obsessive-compulsive disorder (OCD). In addition, we included drugs that are already in use but whose recent approval concerns new modes of administration (long-acting, subcutaneous injection) and other molecules that have not yet been approved but show promising initial evidence in psychiatric therapy. It should be noted that no new pharmacological treatments have been approved recently for anxiety disorders and OCD; as a result, the corresponding sections are significantly less developed compared with those addressing psychotic and mood disorders.

## 2. Methods

In January 2025, an electronic search was performed on PubMed to identify novel pharmacological agents investigated for the treatment of psychiatric disorders, including schizophrenia, BD, MDD, anxiety disorders, and obsessive-compulsive disorder. The following search string was employed: ((“Antidepressive Agents”[Mesh] OR “Adrenergic Uptake Inhibitors”[Mesh] OR “Fluvoxamine”[Mesh] OR “Bupropion”[Mesh] OR “Citalopram”[Mesh] OR “Psychotropic Drugs”[Mesh] OR “Monoamine Oxidase Inhibitors”[Mesh] OR “Dibenzocycloheptenes”[Mesh] OR “Tranquilizing Agents”[Mesh] OR “Phenothiazines”[Mesh] OR “Penfluridol”[Mesh] OR “Serotonin Agents”[Mesh] OR “Dietary Supplements”[Mesh] OR “Antibodies, Monoclonal”[Mesh] OR “Immunotherapy”[Mesh] OR “Liraglutide”[Mesh] OR “semaglutide” [Supplementary Concept] OR antidepressant*[ti] OR antidepressant-agent*[ti] OR Monoamine-Oxidase-Inhibitor*[ti] OR MAO-inhibitor*[ti] OR Adrenergic-Reuptake-Inhibitor*[tiab] OR antimanic[ti] OR Fluvoxamine[tiab] OR bupropion[ti] OR Amfebutamone[ti] OR citalopram[ti] OR psychotropic-drug*[ti] OR psychotropic-agent*[ti] OR psychoactive-drug*[ti] OR psychoactive-agent*[ti] OR Dibenzocycloheptenes[ti] OR tranquilizing-agent*[ti] OR tranquilizing-drug*[ti] OR anti-anxiety-agent*[ti] OR anxiolytic*[ti] OR antipsychotic-agent*[ti] OR neuroleptic*[ti] OR Phenothiazines[ti] OR Penfluridol[ti] OR serotonin-agent*[ti] OR serotonin-drug*[ti] OR dietary-supplement*[ti] OR food-supplement*[ti] OR nutraceutical*[ti] OR Monoclonal-Antibod*[ti] OR Immunotherap*[ti] OR immunization*[ti] OR TV-46000[ti] OR Liraglutide[ti] OR NN2211[ti] OR semaglutide [ti]) AND (“Mood Disorders”[Mesh] OR “Personality Disorders”[Mesh] OR “Anxiety Disorders”[Mesh] OR “Feeding and Eating Disorders”[Mesh] OR “Bulimia”[Mesh] OR “Suicide”[Mesh] AND “Schizophrenia Spectrum and Other Psychotic Disorders”[Mesh] OR mood-disorder*[ti] OR affective-disorder*[ti] OR bipolar[ti] OR Cyclothymi*[ti] OR depressi*[ti] OR Dysthymi*[ti] OR premenstrual[ti] OR SAD[ti] OR seasonal-affective-disorder*[ti] OR seasonal-mood-disorder* OR borderline[ti] OR compulsive[ti] OR schizoid[ti] OR schizotypal[ti] OR anxiety-disorder*[ti] OR neuros*[ti] OR neurotic[ti] OR obsessive-compulsive[ti] OR Anankastic-personalit*[ti] OR Excoriation-disorder*[ti] OR skin-picking-disorder*[ti] OR hoarding*[ti] OR Trichotillomania*[ti] OR Hair-pulling-disorder*[ti] OR panic[ti] OR phobic-disorder*[ti] OR agoraphobia*[ti] OR social-phobia*[ti] OR social-anxiet*[ti] OR sociophobia*[ti] OR feeding-disorder*[ti] OR eating-disorder*[ti] OR anorexia-nervosa[ti] OR Avoidant-restrictive-food-intake-disorder*[ti] OR Food-neophobia*[ti] OR binge-eating*[ti] OR bulimia-nervosa[ti] OR food-addiction*[ti] OR night-eating-syndrome*[ti] OR orthorexia-nervosa[ti] OR Pica[ti] OR rumination-syndrome*[ti] OR suicide*[ti] OR Schizophreni*[ti] OR Psychotic-disorder*[ti] OR psychos*[ti] OR capgras[ti] OR paranoi*[ti] OR Schizoaffective[ti] OR catatoni*[ti] OR Shared-paranoid-disorder*[ti] OR Shared-psychotic-disorder*[ti] OR Folie-a-Deux[ti]) NOT (“Animals”[Mesh] OR animal*) AND (y_10[Filter]). We included the following types of publications from January 2018 to January 2025:-Randomized controlled trials (RCTs),-Open-label trials,-Post hoc subgroup analyses,-Clinical trials,-Retrospective analyses,-Observational studies,-Exploratory analyses,-Narrative reviews,-Systematic reviews,-Meta-analyses.

We excluded case reports, longitudinal studies, prospective studies, and letters to the author. Overlapping studies were also excluded. Eligibility status for articles was determined in the following way: (1) all studies were screened based on the title and abstract, and (2) papers that passed the initial screening were reviewed based on a careful examination of the full manuscript content. In addition, studies in which the diagnosis was not clearly stated, in which severe medical conditions were comorbid, or in which the dosage of the newly administered drug was unclear were excluded. We excluded studies in which the inclusion criteria were not well defined and/or the outcomes were not assessed by self-/hetero-administered scales. RCTs in which the randomization method was not explicit were excluded. The review considered only articles written in English. Details of the studies of newly approved drugs are given in Table 1, Table 2 and Table 3. Drugs that are not listed in the table have not been approved by FDA/EMA for psychiatric disorders but are being studied to assess their efficacy and tolerability.

## 3. Results

The flowchart below illustrates the number of records retrieved from PubMed and the subsequent study selection process (Figure 1).

## 4. Discussion

### 4.1. Schizophrenia

Schizophrenia is a chronic psychiatric disorder that places a substantial physical, social, and economic burden on affected individuals [38]. The disease has a lifetime prevalence of about 1% globally [39]. Relapse prevention is an important goal of maintenance treatment in patients with schizophrenia or other psychotic disorders. Relapses can lead to hospitalization/re-hospitalization, slow and incomplete recovery, progressive cognitive and functional decline, treatment-resistant illness, and reduced ability to maintain employment or relationships [40]. Non-adherence to antipsychotic medication represents the most prevalent and strongly correlated predictor of relapse in individuals with schizophrenia [41]. Antipsychotics are a key part of the treatment of acute episodes of schizophrenia and maintenance pharmacotherapy. The first drugs that were used to treat psychosis were dopamine D2 receptor antagonists, and over time, the dopaminergic pathway has been shown to be strongly involved in this disorder [42]. The most recent drugs (second-generation and third-generation antipsychotics) are characterized by an extended receptor profile, including DA and 5-HT receptor (5-HT) subtypes and significant partial agonism at the D2/D3 and 5-HT1A receptors. The 5-HT1A receptor stimulation leads to the inhibition of 5-HT release and subsequent inhibition of DA release in the prefrontal cortex. A reduction in DA levels in the prefrontal cortex is implicated in the pathophysiology of negative symptoms in schizophrenia. It is hypothesized that partial agonists of the 5-HT1A receptor may increase DA levels in the prefrontal cortex with less negative impact on negative symptoms than atypical antipsychotics [43]. Moreover, 5-HT2C agonism may result in antipsychotic effects without inducing extrapyramidal symptoms [44]. Promising molecular targets include the D3, metabotropic glutamate, 5-HT6, and 5-HT7 receptors. For several decades, the dopaminergic theory contributed to the discovery of typical and atypical antipsychotics and was the only hypothesis for the pathophysiology of schizophrenia. However, except for clozapine, neither typical nor atypical antipsychotics have demonstrated substantial efficacy in alleviating the negative symptoms and cognitive deficits of schizophrenia, which are key determinants of long-term prognosis and functional disability. Of note is the long-acting injectable (LAI) formulation in the treatment of schizophrenia. It is associated with reduced relapse and hospitalization rates compared with daily oral antipsychotics and improved treatment adherence and quality of life [45,46,47,48,49,50]. We have identified several significant developments in the treatment of schizophrenia: the availability of a new antipsychotic, lumateperone; the introduction of new methods of administration, long-acting subcutaneous and intramuscular risperidone and transdermal asenapine; and the introduction of new approaches, such as olanzapine/samidorphan and KarXT. In addition, two other promising molecules have been studied in recent trials but have not yet been approved: ulotaront and ruloperidone. Ulotaront is an agent that acts as a ”trace amine-associated receptor 1” (TAAR1) agonist with 5-HT1A agonist activity, showing promising but preliminary results in the treatment of schizophrenia, free of extrapyramidal symptoms or metabolic side effects [51]. Ruloperidone acts by blocking serotonin, δ-, and α-adrenergic receptors involved in the regulation of mood, cognition, sleep, and anxiety. Its specific therapeutic goal is to address negative symptoms in schizophrenic patients [52].

#### 4.1.1. Lumateperone

Lumateperone (ITI-007) is a once-daily oral atypical antipsychotic that was approved for adults by the FDA in December 2019, both for the treatment of depressive episodes associated with bipolar I or II disorder (as monotherapy and as adjunctive therapy with lithium or valproate) and for schizophrenia [53,54] at a dosage of 42 mg/day. It is an antagonist with high binding affinity to serotonin 5-HT2A receptors, an antagonist with moderate binding affinity to postsynaptic D2 receptors, and a serotonin reuptake inhibitor. Lumateperone also functions as a partial agonist at D1 dopamine receptors, exhibiting moderate binding affinity. This activity may facilitate the indirect modulation of glutamatergic neurotransmission via AMPA and NMDA receptor pathways. Additionally, it demonstrates partial agonism at presynaptic D2 receptors, with a preferential action localized to the mesolimbic system. Notably, lumateperone exhibits minimal activity in the nigrostriatal pathway, thereby conferring a reduced liability for extrapyramidal side effects.

Two randomized, placebo-controlled, double-blind clinical trials were conducted in 334 (aged 18–55) and 450 (aged 18–60) adult patients who met the criteria for schizophrenia or acute psychosis (DSM-5) [6,7]. The dosage of lumateperone ranged from 28 to 42 to 84 mg/day. The primary objective of the studies was to assess the changes in the total Positive and Negative Syndrome Scale (PANSS) score from baseline to the end of the treatment period. Results showed that, after 4 weeks of treatment, lumateperone was associated with a statistically significant improvement in PANSS total score compared with placebo in both studies. In the study by Correll et al., the PANSS positive subscale also improved [7]. The greatest efficacy was reached with a dose of 42 mg/day (at twice the dose, the benefits were comparable) [6,7]. Lumateperone was also associated with significant improvements in other secondary endpoints, including the severity of global symptoms (Clinical Global Impression—Severity (CGI-S) scale) and the global functioning (Personal and Social Performance (PSP) scale) [7]. Lumateperone was generally well tolerated, with no clinically meaningful motor-related adverse effects or significant alterations in cardiometabolic or endocrine parameters observed in comparison with placebo [6,7,53,54]. An open-label study examined the short-term safety/tolerability of lumateperone in 301 outpatients with stable schizophrenia who switched from previous antipsychotics to lumateperone 42 mg once daily for 6 weeks. Following six weeks of treatment with lumateperone, patients were transitioned back to their prior antipsychotic regimen or an alternative antipsychotic. During the lumateperone treatment period, PANSS total scores remained stable [8]. The most common adverse reactions were drowsiness/sedation, dizziness, nausea, constipation, and dry mouth [7,28]. Safety and tolerability of this medication were also suggested by a post hoc study of 1073 patients with acute exacerbation of schizophrenia randomized to receive lumateperone 42 mg/day, risperidone 4 mg/day, or placebo. Adverse events were mild, and discontinuation rates for side effects with lumateperone 42 mg were similar to placebo and lower than with risperidone 4 mg. The only adverse events that occurred at a rate of ≥ 5%, and twice that of placebo for lumateperone, were somnolence/sedation and dry mouth. In addition, the mean change from the baseline in metabolic parameters, prolactin, weight, and extrapyramidal symptoms was similar or reduced in lumateperone 42 mg/day compared with placebo-treated patients and was lower than in risperidone 4 mg/day [9]. Based on the pharmacodynamic profile and clinical data of this medication, depressive symptoms, negative symptoms, and cognition could be the specific domains of action of lumateperone. The peculiar pharmacological mechanisms of action of this drug appear to confer antipsychotic efficacy with favorable safety and tolerability [5].

#### 4.1.2. Risperidone Long-Acting Injectable

Risperidone is a well-known antipsychotic that acts as a D2 and 5-HT2A receptor antagonist. The recommended daily dose is between 2 and 6 mg. It is administered orally, intramuscularly, and subcutaneously, including long-acting formulations. The FDA approved risperidone for schizophrenia in 1993, and the first LAI formulations (every two or four weeks) were approved in 2003 and 2007, respectively.

An alternative method of administration of risperidone LAI was approved by the FDA in July 2018: subcutaneous. It is administered monthly and is indicated for the treatment of schizophrenia in adult patients. One of the main advantages of subcutaneous injection is that it avoids damage to muscle tissue, which is not a rare adverse effect when administering LAI drugs intramuscularly.

The pivotal phase III of the clinical trial randomized 354 inpatients with schizophrenia to receive either 90 or 120 mg of subcutaneous risperidone or subcutaneous placebo on day 1 and day 29. With both dosages, there was a significant improvement in PANSS total score, PANSS Positive Subscale, CGI-S [55], and in health-related quality of life. In a similar way, physical functioning, social integration, and subjective well-being (measured by the scale Subjective Well-being under Neuroleptic Treatment—Short Version, SWN-S) improved. Patient-reported outcomes indicated greater overall treatment satisfaction with the investigational agent relative to placebo and to prior therapeutic formulations [56]. A 52-week open-label phase III study enrolled 408 new, stable patients and 92 rollover participants from Nasser’s RCT [55]. All received 13 monthly subcutaneous injections of risperidone at a dose of 120 mg [10]. PANSS scores continued to improve in the patient rollover from the RCT and remained stable in the new participants. Quality of life remained stable throughout, as did subjective well-being. Satisfaction with the subcutaneous injection increased from week 4 to the end of the study [11]. The most common adverse events in the study, including patients with acute psychosis, were pain at the injection site, constipation, sedation/somnolence, weight gain, and pain in the extremities [55]. In the 52-week safety study, 73.4% of patients reported at least 1 adverse event; the most common were pain at the injection site and weight gain. No changes in vital parameters, laboratory values, or ECG were observed [10].

In 2023, the FDA approved TV46000, which is a long-acting subcutaneous antipsychotic formulation that combines risperidone and an innovative copolymer-based drug delivery technology in a suspension suitable for subcutaneous use [13,57]. This formulation is administered every four weeks (q1m) or every eight weeks (q2m). The approval was based on the results of two randomized phase III clinical trials (RISE study and SHINE study). In both studies, clinical stabilization was initially achieved by oral risperidone therapy. The RISE study was a double-blind trial in which 543 adult patients with schizophrenia were administered TV46000 q1m, TV46000 q2m, or placebo. The results showed that the clinical stabilization achieved with oral therapy was maintained and further improved after administration of TV46000 in both formulations. The findings of the SHINE study were substantially in line with those of the RISE study. This trial consisted of the administration of TV46000 q1m or q2m for 56 weeks in 336 patients suffering from schizophrenia. Frequent adverse events were injection site pain [q1m, 5%; q2m, 4%] and injection site nodules [q1m, 2%; q2m, 6%]. The rates of patients with serious adverse effects (such as worsening symptoms related to schizophrenia and hyperglycemia) were 5% for TV-46000 q1m and 7% for TV-46000 q2m. Eight patients discontinued treatment due to side effects, and no patient had a serious reaction. Therefore, the results of this long-term safety study add to the favorable safety profiles of TV-46000 q1m and q2m, consistent with other risperidone formulations [13].

With regard to the intramuscular administration, in 2022, the EMA approved a new LAI formulation of risperidone. Before 2022, the frequency of risperidone LAI was every 14 days; the new formulation allows the administration every 28 days. We found two studies that preceded its approval [14,58]. In the first multicenter, randomized, double-blind, placebo-controlled 12-month study, 438 patients with schizophrenia received monthly intramuscular injections of risperidone (75 or 100 mg) or placebo for 12 weeks. Results showed a rapid and progressive reduction in symptoms in patients with acute exacerbation of schizophrenia, without the need for supplementation with oral risperidone or loading doses. Moreover, both 75 mg and 100 mg doses were safe and effective, with significant improvements in both PANSS and CGI-S [58]. The most frequently reported side effects were increased prolactin in the blood (7.8%), headache (7.3%), hyperprolactinemia (5%), and weight gain (4.8%). Both doses were well tolerated. The second randomized, double-blind study was conducted on 215 patients with schizophrenia who were either stable (i.e., patients enrolled de novo, with stable symptoms), unstable (i.e., patients who had received a placebo in the previous study [58]), or stabilized (i.e., patients who had already received risperidone intramuscularly monthly in Correll’s study), who received risperidone monthly at a dosage of 75 or 100 mg, again for 52 weeks. The PANSS total score and PANSS positive and negative subscale scores improved in the three groups; CGI-S and CGI-I also improved, especially in the unstable and stabilized patients. Adverse events were headache, hyperprolactinemia, asthenia, weight gain, insomnia, akathisia, and pain at the inoculation site [14].

#### 4.1.3. Transdermal Asenapine

Asenapine is a tertiary amine classified within the dibenzo-oxepinopyrrole chemical group [59]. It is rapidly metabolized in the liver by direct glucuronidation and oxidation [60] with hepatic first-pass metabolism of the oral dose [61]. Initially, a sublingual preparation was developed to bypass hepatic metabolism, and it had a bioavailability of 35% [59]. Transdermal asenapine (HP-3070) was approved by the FDA in 2019 for adults with schizophrenia [15]. Sublingual asenapine is absorbed very quickly, whereas the transdermal formulation is much slower. Transdermal asenapine is associated with a more constant and sustained release that is not influenced by food or drink [59,62]. Three transdermal patch dosages are available (3.8 mg, 5.7 mg, and 7.6 mg per 24 h) corresponding to sublingual doses of 10 mg, 15 mg, and 20 mg per day, respectively. The patch may be applied to various anatomical sites, including the abdomen, upper back, hips, or arms. The study that led to the FDA approval of this formulation was a 6-week phase III RCT on 607 patients with schizophrenia. Patients were randomized to 7.6 mg/24 h, 3.8 mg/24 h, or placebo. There was a significant improvement in PANSS and CGI-S scores [15]. Transdermal asenapine was well tolerated, and the most frequent adverse events were drowsiness, erythema at the application site, dizziness, headache, insomnia, and fatigue [15,62]. It is important to note that transdermal versus sublingual administration could help improve treatment adherence in patients with psychosis.

#### 4.1.4. Olanzapine/Samidorphan

Olanzapine/samidorphan (OLZ/SAM) is a new drug combination that was approved by the FDA in 2021 for the treatment of schizophrenia and BD type I. Olanzapine is a well-known atypical antipsychotic, widely used for the treatment of schizophrenia and BD type I (approved in 1996 by the FDA and in 1997 by the EMA). Samidorphan is a novel pharmacological agent that functions as a mu-opioid receptor antagonist, a receptor implicated in the modulation of mood and behavior. It is thought that samidorphan probably downregulates the endogenous opioid system indirectly enhanced by olanzapine, thus reducing hyperphagia and gratification from food intake. The combination product OLZ/SAM received FDA approval following clinical trials that demonstrated its efficacy both as monotherapy and as adjunctive therapy to lithium or valproate for the treatment of schizophrenia and BD I, respectively. The efficacy of OLZ/SAM in the treatment of schizophrenia was evaluated through six RCTs with placebo or single OLZ [16,17,20,21,23] and four open-label studies [19,22,23]. In these studies, the dosages of both OLZ and SAM ranged from 5 to 30 mg/day. The trial duration was between 3 and 52 weeks. The samples ranged from 36 to 561 patients with schizophrenia. In patients who received OLZ/SAM, the PANSS and CGI-S values remained stable [24] or improved [20] compared with those who received single OLZ [24] or placebo [20,24]. The antipsychotic efficacy of combination treatment (OLZ/SAM) was comparable to that of OLZ monotherapy [21]. Furthermore, the metabolic profile improved in patients treated with OLZ/SAM in terms of weight loss [16,17,19,21,22]. In one study [21], weight gain was registered with OLZ/SAM until week 6 (then the values were stabilized), while with OLZ in monotherapy, the weight gain continued until week 24. Many studies concluded that OLZ/SAM offers a reduction in weight gain, but they could not demonstrate a significant change in metabolic parameters, including insulin, glucose, hemoglobin A1c, triglycerides, and other lipids [21,63]. The OLZ/SAM combination was generally well tolerated, with a safety profile like that of OLZ plus placebo [19,20,21,23]. The main side effects found in the studies examined were drowsiness/sedation, headache, nausea, and dry mouth. The drug does not alter QTc values [23]. The most common adverse events reported were drowsiness [20,22], dry mouth, and suicidal ideation [20]. OLZ/SAM represents a promising therapeutic combination, offering a novel treatment option that may mitigate the risk of weight gain, a common adverse effect associated with OLZ monotherapy.

#### 4.1.5. KarXT

KarXT is a very recent medication that was approved in 2024 by the FDA for the treatment of schizophrenia. It consists of a composition of xanomelin, a muscarinic agonist with an M1/M4 receptor attraction, and trospium, a non-selective muscarinic antagonist that serves to reduce the peripheral cholinergic side effects associated with xanomelin by not penetrating the CNS. M1 and M4 receptor activity indirectly impacts the modulation of dopamine neurotransmission in several brain regions, contributing to several symptoms of severe neuropsychiatric conditions, such as the positive and negative symptoms of schizophrenia and the cognitive deficits and psychotic symptoms in Alzheimer’s disease [26]. We found three trials that assessed the efficacy and tolerability of this new compound [25,26,27]. Sample sizes ranged between 179 and 256 patients with schizophrenia, and the duration of the trials was 5 weeks. Authors administered KarXT (variable dosage ranging from 50 to 125 mg xanomelin and 20 to 30 mg trospium) or placebo twice daily. Patients treated with KarXT showed significant changes in the PANSS total score [25,27], in the PANSS positive subscale score [25], and in the CGI-S compared with patients who received the placebo [27]. The most common adverse events with KarXT were constipation, dyspepsia, headache, nausea, vomiting, hypertension, dizziness, gastroesophageal reflux disease, and diarrhea [25,27]. The rates of extrapyramidal adverse events were similar between the KarXT and the placebo groups, as were the discontinuation rates related to adverse events [27]. These initial results supported the potential of KarXT as a new class of efficacious and well-tolerated antipsychotic drugs based on muscarinic receptor activation and not on the dopamine D2 receptor blocking mechanism.

As mentioned at the beginning of this section, let us now consider two molecules that have not yet been approved, but whose results look promising: ulotaront and ruloperidone.

One noteworthy agent is **ulotaront** (SEP-363856), acting as a TAAR1 agonist with 5-HT1A agonist activity. This new compound showed promising results in the treatment of schizophrenia. In a 4-week, randomized, placebo-controlled study including 245 patients with acute exacerbation of schizophrenia, SEP-363856 (50 mg or 75 mg) was administered to 120 patients. It was found superior to placebo in reducing the PANSS total score. Adverse events included drowsiness and gastrointestinal symptoms. Moreover, one sudden cardiac death occurred in the SEP-363856 group. The incidence of extrapyramidal symptoms and changes in lipid, glycated hemoglobin, and prolactin levels was similar in the study groups [51]. Studies with a longer duration and larger sample are needed to confirm the effects and tolerability of SEP-363856 as well as its efficacy compared with existing pharmacological treatments for patients with schizophrenia. Recognizing its potential, the FDA granted ulotaront Breakthrough Therapy designation for the treatment of schizophrenia in 2019.

**Roluperidone**, currently under study (phase III in the USA), acts by blocking serotonin, δ-, and α-adrenergic receptors involved in the regulation of mood, cognition, sleep, and anxiety. The therapeutic target of roluperidone is to address negative symptoms in patients with schizophrenia. Negative symptoms significantly impact the daily functioning of patients with schizophrenia and unfortunately are relatively unresponsive to antipsychotics with a dopaminergic mechanism. It is therefore necessary to identify new molecules that may prove effective on this symptom dimension.

Two RCTs on the efficacy of this agent are available [52,64], respectively, in samples of 244 and 513 patients with schizophrenia. In both studies, patients received roluperidone 32 mg/day, roluperidone 64 mg/day, or a placebo for 12 weeks. The authors stated that the Negative Symptom Factor Score (NSFS), PANSS total score, PANSS negative subscale score, and CGI-S score improved significantly in the group that received roluperidone at a dose of 64 mg compared with placebo. The most common adverse effects, of mild to moderate severity, were insomnia, worsening schizophrenia, headache, anxiety, and agitation. Two deaths were registered among patients who received roluperidone, but they were not treatment related [52].

These new agents are expected to produce different responses from the conventional pharmacological paradigm, potentially proving effective in patients previously considered resistant to pharmacotherapy [5].

In recent years, several other molecules have been studied to improve the treatment of schizophrenia-related symptoms: anti-inflammatory drugs, serine, sarcosine, glycine, and evenamide. Nevertheless, the results are still sparse or do not reach statistical significance and LU AF11167 [65,66].

The results are summarized in Table 1.

### 4.2. Bipolar Disorders

BD affects between 1% and 5% of the world’s population and is characterized by recurrent episodes of mood alterations, oscillating between mania or hypomania and depression, accompanied by changes in energy and activity levels. Its causes are not yet fully understood, but it derives from the interaction of genetic, neurobiological, and environmental factors [67]. The dopaminergic, serotonergic, GABAergic, and glutamatergic systems are key targets in the pharmacological treatment of BD, both for symptom control during acute phases and for mood stabilization. Treatment includes the acute management of manic or hypomanic episodes, maintenance therapy to prevent relapses and new episodes, and treatment for bipolar depression. Approximately one-third of patients do not respond to at least two treatment options and are considered treatment resistant. Suicide is a severe complication of BD, closely linked to depressive and mixed episodes [68].

In recent years, a new antipsychotic, lumateperone, and a new formulation of olanzapine combined with samidorphan, a molecule capable of reducing weight gain in patients undergoing treatment, have been approved. The mechanisms of action of these two drugs are discussed in more detail in the section on schizophrenia. The results are summarized in Table 2.

#### 4.2.1. Lumateperone

Regarding bipolar depression, two independent clinical studies were conducted to evaluate the efficacy of lumateperone both as monotherapy and as adjunctive therapy to lithium or valproic acid in adult patients meeting DSM-5 criteria for depressive episodes associated with BD I or BD II. The primary efficacy outcome in both studies was the change from baseline in the total Montgomery–Åsberg Depression Rating Scale (MADRS) score at week 6.

As monotherapy, the efficacy of lumateperone was assessed in a multicenter, randomized, double-blind, placebo-controlled study lasting 6 weeks, which enrolled 377 adult patients (mean age: 45 years) with bipolar depression. Lumateperone demonstrated a statistically significant improvement in depressive symptoms, with a mean reduction of 16.8 points in MADRS scores compared with a 13.5-point reduction in the placebo group [28]. Additionally, the efficacy of lumateperone as an adjunctive therapy to lithium or valproate was evaluated in another multicenter, randomized, double-blind, placebo-controlled study of six weeks’ duration. This study enrolled 528 adult patients (mean age: 46 years) experiencing a major depressive episode while already receiving stable doses of lithium or valproate as their primary mood stabilizer. The results showed a dose-dependent trend toward improvement in depressive symptoms, with a mean reduction of 11.2 points in the total MADRS score for the lumateperone group compared with a reduction of 6.6 points in the placebo group. Lumateperone was also associated with statistically significant improvements in other secondary endpoints in both studies, including scores on the CGI-S scale and the Sheehan Disability Scale (SDS). Adverse events were generally mild to moderate and consistent with those observed in previous studies on schizophrenia, with no new adverse effects reported. These data further support the favorable safety and tolerability profile of lumateperone across varying psychiatric populations [29]. However, when lumateperone was taken in combination with lithium or valproate, the incidence of adverse effects increased. In particular, drowsiness, dizziness, and nausea were more frequently observed in the lumateperone group. Only two cases of mild akathisia were reported [29].

#### 4.2.2. Olanzapine/Samidorphan

The use of olanzapine was approved in 2000 for the treatment of acute or mixed manic episodes, in 2003 for the treatment of bipolar depression in combination with fluoxetine, and in 2004 for relapse prevention (FDA). In 2021, the OLZ/SAM combination was approved for short-term (acute) and maintenance treatment of manic or mixed episodes associated with bipolar I disorder. It was also approved in combination with valproate or lithium for the treatment of manic or mixed episodes in bipolar I disorder.

The rationale for the OLZ/SAM combination is to mitigate weight gain while preserving the antipsychotic efficacy of olanzapine [53]. A study conducted by Kahn et al. [24] assessed the effectiveness of this combination in reducing weight gain compared with olanzapine alone. The study enrolled 428 patients diagnosed according to DSM-5 criteria with schizophrenia (268), schizophreniform disorder (66), or bipolar I disorder (92), aged between 16 and 39 years. Patients were randomized to receive OLZ/SAM (5–20/10 mg/day) or olanzapine (5–20 mg/day). A total of 33.1% of patients in the OLZ/SAM group experienced a ≥ 7% weight gain compared with 44.8% in the olanzapine group. The change in waist circumference was 2.99 cm vs. 3.90 cm, respectively. Furthermore, a negative change in the CGI-S score was observed in patients treated with OLZ/SAM, reflecting an improvement in their clinical condition. Additionally, OLZ/SAM and olanzapine demonstrated similar safety profiles. Weight gain was more pronounced in patients receiving olanzapine as early as week 6, and this trend persisted through the assessment at week 12.

Recently, numerous studies have investigated the potential use of ketamine and esketamine in alleviating suicidal ideation and depressive symptoms through NMDA receptor antagonism. Recent investigations also suggested that BD is associated with immune system dysfunction, characterized by increased proinflammatory cytokines in the nervous system of affected individuals [69]. The hypothesis that the immune system plays a key role in BD progression was initially proposed based on the potential immunomodulatory effects of lithium in mood stabilization. Indeed, both manic and depressive episodes in BD are associated with the activation of neuroinflammatory pathways, as indicated by increased acute-phase proteins and pro-inflammatory cytokines [67]. For example, manic states in BD are characterized by elevated levels of acute-phase proteins, including interleukin (IL), tumor necrosis factor-α (TNF-α), and interferon-γ (IFN-γ) [69], as well as haptoglobin, fibrinogen, and C-reactive protein (CRP), with a particularly pronounced increase in CRP during manic phases [67,70]. Furthermore, BD patients exhibit a higher incidence of multisystem inflammatory diseases. The immune-inflammatory hypothesis opens new therapeutic perspectives for BD, including the opportunity to develop novel pharmacological approaches and to optimize targeted therapeutic strategies using monoclonal antibodies [69]. Based on these new approaches, numerous studies have been conducted to evaluate the potential efficacy of the new molecules in BD. These medications are not currently approved for the treatment of BD. However, we have chosen to include them in this review because research in recent years has focused on the use of these molecules in the setting of bipolar depression.

#### 4.2.3. Ketamine and Esketamine

Ketamine and esketamine are used in the treatment of treatment-resistant bipolar depression (TR-BD). Compounds that modulate glutamatergic transmission are considered effective in alleviating symptoms of depressive episodes in BD as well as symptoms related to mixed conditions [71]. The principal pharmacological target of ketamine is the antagonism of ionotropic NMDA receptors (NMDARs). At subanesthetic doses, ketamine increases glutamate release in the medial prefrontal cortex (mPFC) by selectively inhibiting NMDARs on GABAergic interneurons, thereby disinhibiting glutamatergic transmission from pyramidal neurons in the mPFC. Excess extracellular glutamate activates synaptic AMPA receptors, leading to increased release of brain-derived neurotrophic factor (BDNF). BDNF promotes synaptic protein synthesis, dendritic spine formation, and synaptic strengthening, reversing stress-induced deficits in synaptic plasticity.

Furthermore, ketamine interacts with other neurotransmitter systems, including dopaminergic, serotonergic, adrenergic, and cholinergic pathways [72], as well as neuroinflammatory mechanisms such as the kynurenine pathway [73]. In addition to its glutamatergic effects, ketamine and esketamine stabilize neural cell membranes by modulating the tonic influx of Ca^2+^ and Na^+^.

Given the large body of literature on this topic, we selected the most recent studies with the greatest clinical relevance. We reviewed four open-label studies that enrolled between 13 and 53 patients with TR-BD, treated with intravenous ketamine at a dose of 0.5 mg/kg, administered from a minimum of 1 to a maximum of 8 times. The results indicated that although ketamine is not yet approved for the treatment of TR-BD, it represents a promising adjunctive treatment for BD, with a good overall response rate. Additionally, its anti-suicidal effect was particularly notable [74,75,76]. The response rate in the three studies ranged from 24.5% to 73%. Improvement in depressive symptoms was assessed using the Hamilton Depression Rating Scale (HDRS) [75], the MADRS, and the Scale for Suicidal Ideation (SSI) [74,76]. In general, the treatment was well tolerated, with no occurrence of serious adverse events.

One open-label study conducted by Zhuo et al. [77] investigated the effects of intravenous ketamine infusion over 3 weeks in 38 patients with bipolar depression. Participants received multiple infusions (nine in total). A significant reduction in depressive symptoms was observed after the first three doses, as assessed by HDRS, followed by a subsequent worsening of symptoms and a full relapse by the third week. This discrepancy could be attributed to the small sample size or a higher level of treatment resistance in the study group. Additionally, all patients received antidepressants alongside mood stabilizers, which may have influenced the results [68].

A retrospective analysis conducted by McIntyre et al. [71] on 201 patients with treatment-resistant MDD or BD demonstrated that ketamine rapidly reduced refractory symptoms such as anxiety, irritability, and agitation (AIA), with a particularly notable effect on anhedonia. Among these patients, 113 exhibited mixed characteristics. AIA symptoms were measured using the Generalized Anxiety Disorder 7-item scale (GAD-7). The overall treatment response was assessed using the Quick Inventory for Depressive Symptomatology Self-Report-16 (QIDS-SR16), while changes in suicidal ideation were evaluated using the specific suicide item of the QIDS-SR16. Patients with AIA experienced a significantly greater reduction in overall depressive symptoms, suicidal ideation, anxiety, irritability, and agitation [71]. A single-arm open-label observational study by Fancy et al. [78] assessed the real-world effectiveness of repeated ketamine infusions for individuals with treatment-resistant bipolar depression. A total of 66 patients diagnosed with BD received intravenous ketamine at an initial dose of 0.5 mg/kg diluted in 0.9% saline solution and infused over 40 min. If the response was inadequate, the dosage was increased to 0.75 mg/kg for the third and fourth infusions. Patients received 4 infusions over 8–14 days, depending on their availability and scheduling. A post-treatment evaluation was conducted one week after the fourth infusion, assessing improvement using the QIDS-SR16. The study found significant reductions in suicidal thoughts (QIDS-SR16 suicide item), GAD-7, and functional impairment (Sheehan Disability Scale). After four intravenous infusions, the response rate was 35%, while the remission rate was 20%.

Overall, the literature highlights that ketamine is a valuable intervention for rapidly improving depressive symptoms and suicidal ideation in BD. Additionally, ketamine has demonstrated efficacy in the rapid treatment of AIA in adults with mood disorders resistant to conventional therapies, as well as in alleviating anhedonia, a symptom strongly associated with increased suicide risk. However, intravenous ketamine is currently approved only for the treatment of treatment-resistant depression (TRD), and it remains an off-label option for BD.

The use of esketamine nasal spray (ESK-NS) in the treatment of bipolar depression is supported by two main lines of evidence. First, neuroimaging studies have shown alterations in glutamate neurotransmission at different stages of BD, particularly affecting the dorsolateral prefrontal cortex and the anterior cingulate cortex, two brain regions crucially involved in mood regulation. Second, ketamine, a modulator of the glutamatergic system, has demonstrated good clinical efficacy and tolerability as an antidepressant in the treatment of bipolar depression [79].

ESK-NS has a more favorable safety profile than ketamine, supporting its potential use in outpatient settings. This is further confirmed by the SUSTAIN-2 study, in which no manic symptoms were reported among adverse events [80]. According to the EMA, the use of ESK-NS is not contraindicated in patients with BD, suggesting that treatment decisions should be based on a careful risk–benefit assessment [79].

Most studies on esketamine do not specifically evaluate its efficacy in bipolar depression, but rather include mixed samples of patients with MDD, TRD, and treatment-resistant bipolar depression (BD-TRD). To avoid overlapping with the studies presented in the chapter on depression, we chose to report only one recent study by Santucci et al. [81], which exclusively focused on patients with bipolar depression.

This study assessed the efficacy and safety of ketamine and esketamine in a clinical setting for bipolar depression. A total of 45 patients with bipolar depression were treated with either intravenous ketamine (0.5 mg/kg over 40 min) or intranasal esketamine (56 mg or 84 mg), administered twice weekly for up to 4 weeks. Among participants, 39% achieved a clinical response, and 13.2% achieved remission, based on improvements in MADRS scores. No patients experienced mania or hypomania during the acute treatment phase. However, during the maintenance phase, 28.9% of participants developed symptoms suggestive of hypomanic or manic episodes [81].

#### 4.2.4. NRX-101 (D-Cycloserine Plus Lurasidone)

D-cycloserine (DCS) is both an antibiotic used for tuberculosis and a compound with antidepressant properties [82]. It acts as an NMDA receptor agonist at lower doses and an NMDA antagonist at higher doses. When administered in doses greater than 500 mg/day, DCS has demonstrated antidepressant effects in patients with TRD when used in combination with selective serotonin reuptake inhibitors (SSRIs). It has also been found to enhance the antidepressant effects of ketamine in patients with BD and BD-TRD. Additionally, DCS has been shown to help sustain remission from suicidality following ketamine infusion, particularly in patients with bipolar depression. DCS is considered a favorable oral antidepressant option, as it does not carry the risk of neurotoxicity or abuse potentially associated with direct NMDA channel-blocking agents. Overall, DCS appears to be an effective and safe antidepressant treatment [82]. Nierenberg conducted in 2023 a multicenter, double-blind, two-phase, parallel-randomized study to evaluate whether a fixed-dose combination of DCS and lurasidone (NRX-101) was more effective than lurasidone alone in maintaining improvement following initial treatment with intravenous ketamine in patients with BD, severe depression, and acute suicidal ideation or behavior. A total of 22 patients aged 18–65 years with BD depression and suicidal ideation or behavior who had received either ketamine or saline infusion were enrolled in the study. Those who demonstrated improvement were randomly assigned to treatment with lurasidone or NRX-101. Depressive symptoms were assessed using the MADRS, while suicidal thoughts and behaviors were measured using the Columbia Suicide Severity Rating Scale (C-SSRS) at 28 and 42 days. The fixed-dose combination of DCS and lurasidone was found to be significantly more effective than lurasidone alone in maintaining improvements in depression and reducing suicidal ideation, as assessed by the C-SSRS and CGI-SS. A reduction in akathisia was also observed in patients treated with NRX-101, consistent with results obtained in preclinical studies.

#### 4.2.5. Celecoxib

Celecoxib (CBX) is a selective COX-2 inhibitor that has been investigated as a potential adjunct to first-line mood-stabilizing treatments such as lithium and valproate. Several small studies suggest that the addition of anti-inflammatory agents may improve depressive symptoms in BD.

A study conducted by Edberg et al. [83] examined the inflammatory profile of patients who had participated in previous trials evaluating celecoxib as an adjunctive treatment for BD. This study specifically analyzed blood levels of the pro-inflammatory biomarker CRP. The sample consisted of 47 patients, who were randomized to receive either escitalopram (10 mg twice daily) + CBX (200 mg twice daily) or escitalopram (10 mg twice daily) + placebo. Results showed that baseline CRP levels were significantly higher in BD patients than in healthy controls, suggesting that CRP may serve as a useful biomarker for BD. Although no significant differences in CRP levels were observed between the CBX and placebo groups at baseline, by week 8, CRP levels were significantly reduced in the CBX group compared with placebo. This suggests a reduction in inflammation in CBX-treated patients and highlights CRP as a potential biomarker for monitoring treatment response in patients receiving SSRI + CBX therapy [83].

A more recent RCT by Husain et al. [84] evaluated the efficacy of minocycline (200 mg, an antibiotic with anti-inflammatory properties) + celecoxib (400 mg) versus either agent alone or placebo. The study included 254 patients experiencing a depressive episode associated with BD (type I or II) and lasted 12 weeks. The primary outcome was a reduction in depressive symptoms, assessed using the 17-item Hamilton Depression Rating Scale (HAMD-17). No significant differences in HAMD-17 score reductions were observed between treatment groups and placebo. Additionally, no significant differences in serious adverse events were reported across the groups. These findings cast doubt on the potential therapeutic benefits of adjunctive anti-inflammatory drugs for the acute management of BD.

#### 4.2.6. Infliximab

Infliximab is a TNF inhibitor that acts on a pro-inflammatory cytokine found to be elevated in individuals with BD and potentially linked to cognitive impairments observed in this clinical population [70]. McIntyre et al. [71] conducted an RCT to assess the efficacy of adjunctive infliximab treatment in BD I or II patients with baseline indicators of inflammation. The 12-week study included 60 patients aged 18–65 years who were randomly assigned to receive three intravenous infusions of infliximab or placebo at baseline and at weeks 2 and 6. Depressive symptom severity was assessed using the MADRS, with no significant differences observed between the infliximab and placebo groups at week 12. However, secondary analyses using the Childhood Trauma Questionnaire revealed a significant interaction between treatment, time, and history of childhood maltreatment. Specifically, infliximab-treated patients with a history of physical maltreatment exhibited greater reductions in MADRS total scores and higher response rates (≥ 50% reduction in the MADRS total score) compared with placebo [71]. While infliximab did not significantly reduce depressive symptoms in the overall cohort of adults with bipolar depression, these findings, bearing in mind that these are preliminary data derived from a post hoc analysis, suggest that individuals with a history of physical and/or sexual abuse may benefit from anti-inflammatory treatment. Further studies are warranted to explore the potential role of infliximab in BD patients with inflammation-related depressive symptoms.

### 4.3. Major Depressive Disorder

MDD is a leading cause of disability worldwide, significantly impacting the quality of life and social and economic functioning of individuals. According to the most recent estimates, the annual incidence of MDD is approximately 5–7% of the global population, with a lifetime prevalence ranging between 15% and 20%. The condition predominantly affects women, with a gender ratio of about 2:1 compared with men, and it is often associated with psychiatric and medical comorbidities, including anxiety disorders, substance abuse, and cardiovascular disease. MDD arises from a complex interplay of genetic, biological, psychological, and environmental factors. Key biological hypotheses involve alterations in neurotransmitter systems (serotonin, norepinephrine, and dopamine), dysfunction of the hypothalamic–pituitary–adrenal (HPA) axis, neuroinflammation, and reduced neuroplasticity. Recent studies have highlighted the role of the glutamatergic system and neurotrophic pathways, such as brain-derived neurotrophic factor (BDNF), in the onset and maintenance of depressive symptoms [85]. Glutamatergic-based antidepressants have shown promising benefits compared with conventional SSRIs and SNRIs, including enhanced functional outcomes, more rapid alleviation of symptoms, and accelerated achievement of remission [86]. Research based on these biological mechanisms has led to the development of novel antidepressant treatments such as esketamine, approved by the FDA in 2019 for the treatment of TRD, dextromethorphan/bupropion, approved by the FDA in 2022 for the treatment of MDD, and brexanolone and zuranolone, approved in 2019 and 2023, respectively, for the treatment of postpartum depression. The results are summarized in Table 3.

#### 4.3.1. Ketamine and Esketamine

The mechanisms of action of ketamine and esketamine were discussed in the chapter on BD. The enantiomer esketamine was approved in 2019 for TRD and acute suicidal ideation, whereas ketamine itself is not currently approved for any psychiatric disorder but is used off-label via intravenous (IV) administration for TRD and acute suicidal ideation [87].

As the literature on ketamine and its enantiomer is very extensive, we selected the most recent and relevant studies to provide a broad perspective on the short- and long-term therapeutic potential, safety, and tolerability of these treatments. We reviewed four double-blind, placebo-controlled randomized controlled trials (RCTs) involving IV administration of ketamine [23,88,89] or IV administration of esketamine [90]. These studies included samples ranging from 36 to 156 patients diagnosed with MDD, BD, or TRD with suicidal ideation. The treatment protocols varied, with a minimum of a single infusion of 0.5 mg/kg of ketamine [91] and a maximum of four infusions over two weeks [88]. Participants’ ages ranged from 13 to 76 years. Improvement in suicidal ideation was assessed using the SSI [89,90], the C-SSRS [89,90,91], the Suicide Probability Scale (SPS) [88], the Positive and Negative Suicide Ideation Inventory (PANSI), and the MADRS item 10 score [91]. All studies demonstrated a rapid improvement in suicidal ideation, with greater effects observed in patients diagnosed with BD [89] and those with moderate and low refractoriness (current depressive episode < 24 months or ≤ 4 failed antidepressant treatments) [91].

A phase III, open-label, long-term extension study (SUSTAIN-3) was conducted to evaluate the long-term efficacy of esketamine nasal spray [31]. The sample included 1148 patients aged > 18 years with TRD, of whom 458 completed the 4-week induction phase followed by an optimization/maintenance phase, while 690 discontinued before completing the maintenance phase. In the induction phase, participants received intranasal esketamine (28 mg, 56 mg, or 84 mg) twice weekly for four weeks in a flexible dosing regimen. During the optimization/maintenance phase, dosing intervals were adjusted based on depression severity. All patients were also treated with an oral antidepressant (except MAO inhibitors). The mean cumulative duration of maintenance treatment with esketamine was 31.5 months, with a maximum of 4.5 years. The improvement in depressive symptoms was assessed by changes in the MADRS scores. At the end of the induction phase, 35.6% of participants achieved remission (MADRS score ≤ 12), which increased to 46.1% in the optimization/maintenance phase. The most common adverse events included headache, dizziness, nausea, dissociation, drowsiness, and nasopharyngitis. Regarding suicidal ideation, 4.3% of participants without prior suicidal ideation reported new occurrences during the study, and 10 participants (0.9%) exhibited new suicidal behaviors, 9 of whom had a history of suicidal ideation. An improvement in C-SSRS severity categories from baseline was observed in 14.0% of participants. Overall, 5.6% of participants experienced 1 or more adverse event potentially related to suicidality, with 1 participant dying by suicide [31].

#### 4.3.2. Dextromethorphan/Bupropion

AXS-05, a combination of dextromethorphan and bupropion, is formulated as an extended-release tablet containing 45 mg of dextromethorphan bromide and 105 mg of bupropion hydrochloride. It received FDA approval in 2022 for the treatment of MDD in adults. Dextromethorphan modulates glutamate signaling through non-competitive NMDA receptor antagonism, sigma-1 receptor agonism, and inhibiting serotonin reuptake. Bupropion, on the other hand, increases dextromethorphan bioavailability by inhibiting the CYP2D6 enzyme [92], and it improves monoaminergic function by acting as an inhibitor of norepinephrine and dopamine reuptake [86]. The combination dextromethorphan/bupropion has been shown to be effective in rapidly improving symptoms of depression in patients with MDD, including those with TRD or suicidal ideation. The single-tablet formulation ensures a consistent pharmacokinetic profile and improves adherence compared with multiple treatment regimens, since dextromethorphan achieves a significantly higher exposure compared with non-administration in combination. A phase 3 RCT (GEMINI) was conducted on a sample of 327 patients with MDD with moderate-to-severe major depressive episodes ongoing for at least 4 weeks. The patients enrolled were aged 18–65 years. Patients were randomized into two groups: 163 patients received dextromethorphan/bupropion (45–105 mg tablet) and 164 received placebo (once daily for days 1–3, twice daily thereafter). Improvement in symptoms was assessed by MADRS and CGI-S. Clinical response was obtained by 54.0% with dextromethorphan/bupropion compared with 34.0% with placebo at week 6. Significant improvement in depressive symptoms compared with placebo occurred starting one week after treatment initiation and was generally well tolerated. The most frequent side effects were dizziness, nausea, headache, drowsiness, and dry mouth. Neither psychomimetic effects of the drug nor sexual dysfunction were reported [33]. Two other studies were conducted [92]: COMET, a phase 3, multicenter, open-label US trial, and EVOLVE, an open-label trial to evaluate the long-term efficacy of the drug. Both studies enrolled, respectively, 876 and 186 patients who were already taking or started taking dextromethorphan/bupropion twice daily for a minimum duration of 12 months to a maximum of 15 months. The improvement in depressive symptoms was assessed by MADRS and CGI-I and that of functioning by SDS. Remission was achieved in 46% of participants in the EVOLVE study and 52.5% in the COMET study at week 6. Improvements in MADRS and SDS scores were maintained for one year. The most common adverse events overlapped with adverse effects found in previous studies. An aggregate analysis of the GEMINI and EVOLVE studies showed that AXS-05 notably enhanced depressive symptoms independent of prior antidepressant treatment, ethnicity, and gender [92]. Another randomized, double-blind, clinical trial conducted by Tabuteau et al. [32] evaluated the difference in efficacy between dextromethorphan/bupropion vs. bupropion in a total sample of 80 patients with MDD. MADRS assessed the clinical improvement. Remission rates were significantly higher in the dextromethorphan/bupropion group from week 2 onward, with response rates at week 6 of 60.5% in the dextromethorphan/bupropion group and 40.5% in the bupropion group.

In summary, dextromethorphan/bupropion seems to be a useful intervention for major depressed patients, acting rapidly on depressive symptoms and global functioning, with a long-lasting effect in the absence of severe side effects.

#### 4.3.3. Brexanolone

The FDA approved brexanolone in March 2019 for the treatment of postpartum depression (PPD) in adult women. PPD is a severe condition affecting between 13% and 19% of women who have just gone through motherhood [93]. According to the DSM-5, postpartum depression is not defined as a separate diagnostic category but falls under the diagnosis of MDD with perinatal onset specifier. It applies when depressive symptoms begin during pregnancy or within the first four weeks after delivery. This disorder is characterized by a persistent state of depressed mood in new mothers, accompanied by feelings of sadness, worthlessness, and despondency. Brexanolone is a cyclodextrin-based formulation of the neurosteroid allopregnanolone (AlloP), a potent and effective positive allosteric modulator of GABA-A receptors developed for intravenous administration [34,35]. Two double-blind, randomized, placebo-controlled, phase 3 trials conducted by Meltzer-Brody [35] analyzed the effectiveness of brexanolone in patients with PPD. Study 1 compared the efficacy of administering brexanolone 90 μg/kg per hour (BRX90), brexanolone 60 μg/kg per hour (BRX60), or a placebo; study 2 compared BRX90 with a placebo. Treatment or placebo was administered as an intravenous infusion for 60 h. Efficacy was assessed by the change in the Hamilton Depression Rating Scale (HAM-D) score during the infusion, with a 30-day follow-up. The mean least squares reduction in the total HAM-D score in study 1 compared with the baseline was 19.5 points in the BRX60 group, 17.7 in the BRX90, and 14.0 in the placebo group. In study 2, at 60 h, the mean HAM-D score reduction from baseline was 14.6 points in the BRX90 group and 12.1 points in the placebo group.

The results in the treatment groups were maintained until the end of the 30-day follow-up period and were superior to those in the placebo group. In addition, more patients in the treatment groups achieved remission than in the control group. In study 1, one patient in the BRX60 group had two serious adverse events (suicidal ideation and intentional overdose attempt during follow-up). In study 2, one patient in the BRX90 group had two serious adverse events (altered consciousness and syncope), which were considered treatment related. In the remaining study participants, side effects were minimal and similar to previous studies, with headache, dizziness and drowsiness being the most common symptoms, all considered mild and self-limiting. Brexanolone represents an innovative option for the treatment of moderate-to-severe PPD, providing rapid and effective resolution of symptoms. Limitations to its use include the need for hospitalization, intravenous administration, high costs, and in rare cases, risk of sedation or loss of consciousness. However, the risk–benefit ratio seems favorable for the treatment of moderate-to-severe PPD, with potentially significant benefits for maternal health in the postpartum period [94].

Regarding the use of brexanolone during breastfeeding, a study conducted by Wald et al. [95] investigated this issue. Twelve healthy women who had given birth within the past six months received an intravenous infusion of brexanolone (up to a maximum of 90 mg/kg per hour) for 60 h. The peak allopregnanolone levels in breast milk (125 mcg/L) were recorded between 24 and 48 h after the start of the infusion. Most women exhibited drug levels in breast milk below the detection limit within three days after treatment completion. A pharmacokinetic model predicted that 95% of patients would have allopregnanolone levels in breast milk below 10 mcg/L at 36 h post-infusion. The amount of drug transmitted to the infant through breast milk was found to be very low (median 0.69%, maximum 1.3% of the maternal dose) [95].

However, according to the information available in the brexanolone prescribing label, no data are available on the effects of brexanolone on a breastfed infant. While the existing data on its use during lactation do not suggest a significant risk of adverse reactions in neonates, further studies are needed. Currently, the decision to use brexanolone during breastfeeding should be based on the mother’s clinical condition and the potential adverse events that may occur in the infant (FDA, 2019).

#### 4.3.4. Zuranolone

Zuranolone (SAGE-217) was approved by the FDA in August 2023. It is an investigational neuroactive steroid (NAS) and positive allosteric modulator of GABA-A receptors with a mechanism of action similar to brexanolone and a pharmacokinetic profile suitable for once-daily oral administration [36], overcoming limitations to the use of brexanolone such as the need for hospitalization and intravenous administration. Zuranolone represents a promising therapeutic option in the treatment of PPD. One of its advantages is the short duration of treatment, requiring only two weeks. In addition, its rapid action makes it a potentially more effective option than SSRIs and SNRIs.

Two phase 3, double-blind, randomized, placebo-controlled trials evaluated the primary outcome of treatment with zuranolone vs. placebo. The efficacy of this drug was evaluated in the first study at a dosage of 30 mg/day [36] and in the second trial at 50 mg/day [37]. Zuranolone was taken once daily for 14 days. In the first study, 153 patients were enrolled, and in the second, 196 subjects. In both studies, patients could continue taking antidepressants if they maintained a stable dosage for at least 30 days before the first treatment dose. All patients were evaluated in the post-treatment follow-up until day 45. The following rating scales were used to assess treatment efficacy: HAM-D, HAM-A, CGI-S, MADRS [36], and Patient Health Questionnaire (PHQ-9) [37]. Zuranolone was superior to placebo in the percentage of patients who achieved a reduction in HAM-D score from baseline, with improvements observed as early as day 3 and maintained consistently at subsequent visits [36,37]. Improvements in anxiety symptoms and CGI-I response were also shown up to day 45 [36]. The most common adverse events with zuranolone were drowsiness, dizziness, sedation [37], headache, and diarrhea [36]. No loss of consciousness, withdrawal symptoms, or increased suicidal ideation or behavior were observed.

A recent study conducted by Ahmad [96] collected data from several studies, selecting four RCTs in which zuranolone was used at various dosages vs. placebo in patients with MDD. The total study sample was 1357 patients whose clinical course was assessed with HAM-D, HAM-A, and MADRS. Patients treated with zuranolone showed a statistically significant effect in the score change in all the evaluation scales (as early as day 15 of treatment) compared with the placebo group. Zuranolone was also significantly associated with a higher response and remission rate compared with placebo. Zuranolone is an effective and safe drug for short-term monotherapy of MDD. It shows results in 14 days (compared with 2–4 weeks for SSRIs) and has anti-anxiolytic effects. However, further studies are needed to determine the long-term effects with a larger sample of patients [96].

It is essential to consider the complex and multifactorial nature of postpartum depression. Although drug therapies represent only one aspect of treatment, factors such as cultural context, gender roles, and social expectations profoundly influence maternal mental health. Therefore, an approach that takes these elements into account is crucial to ensure effective interventions. Further research will be needed to evaluate how zuranolone compares with other therapies, its long-term potential, possible new indications for use, and the safest and most effective ways to administer it [97].

#### 4.3.5. Psilocybin

Some studies are available on the therapeutic potential of psychedelics, in particular on their effects on consciousness, emotion processing, mood, and neural plasticity. Regarding MDD, some recent studies have focused on psilocybin, a partial agonist of the 5-HT2A receptor. The psychedelic effects of psilocybin result primarily from the activation of 5-HT2A receptors located in the cortex and subcortical structures. These receptors are concentrated in the apical dendrites of pyramidal neurons in layer 5 of the neocortex, with a high density in the prefrontal cortex (PFC). This distribution is related to psilocybin’s effects on visual perception, attention, and other cognitive processes. Its stimulation is essential to produce the perceptual and cognitive changes typical of psychedelics. Other serotonergic receptors, such as 5-HT1A, are involved in the mechanisms of action of psylocibin. Studies in animal models show that psilocybin can induce neuroplasticity with antidepressant effects similarly to ketamine. Compounds that promote neuronal plasticity are defined as psychoplastogens and may have therapeutic applications in depression [98].

Preliminary studies of psilocybin have shown therapeutic potential through its effects on the neural circuits and key brain regions implicated in depression, including the amygdala [99]. In a phase 2 double-blind trial, the efficacy of psilocybin was tested in TRD [100]. The study sample consisted of 233 patients, who were divided into 3 groups and received psilocybin doses of 25 mg, 10 mg, and 1 mg (control), respectively. Higher doses demonstrated significantly greater efficacy compared with the 1 mg dose on the primary efficacy endpoint as assessed by the MADRS scale. Antidepressant effects were rapid, appearing as early as day 2, and one-fifth of participants who received psilocybin 25 mg/day maintained a response from week 3 to week 12. Adverse events occurred in 77% of participants, the most common being headache, nausea, and dizziness. Serious adverse events were more likely to occur with dose escalation; the same correlation was found for suicidal ideation, suicidal behavior, and self-harm [100]. Another phase II open-label study involved 19 participants who received a single dose of psilocybin (25 mg) with psychological support before and after administration [101]. Psylocibin was administered as adjunctive treatment for TRD patients taking a concomitant SSRI. The aim was to test the compatibility of psilocybin with existing antidepressant therapies in patients who cannot discontinue SSRIs for clinical reasons. Results showed an improvement in depressive symptoms, with a clinically significant reduction in MADRS score at week 3 that was evident from day 2 and maintained over time. In addition, 42.1% of participants met both response and remission criteria. The CGI-S score decreased, indicating a moderate improvement. In terms of safety, psilocybin was not responsible for severe side effects, although they are rather frequent [100]. When more severe adverse events occurred, these receded by reducing the dosage.

### 4.4. Anxiety Disorders

Anxiety disorders are among the most prevalent psychiatric conditions and represent a significant cause of disability. The serotonin system has been extensively investigated in relation to the etiology of these disorders. First-line treatment options for anxiety disorders primarily involve serotonergic agents, including SSRIs, SNRIs, and azapirones such as buspirone. Nevertheless, side effects are common and affect treatment adherence in about 50% of patients [102]. In recent years, new agents acting on different 5-HT receptors have been studied in order to obtain clinical effects (similar to those of SSRIs) and more favorable tolerability profiles. The FDA has approved no new drugs for any anxiety disorders in recent years, and there is a relative lack of new drugs under investigation for these disorders [103]. Among studies on serotonergic (such as agomelatine and vortioxetine), noradrenergic, glutamatergic (such as riluzole and ketamine), and neuropeptides, very few advanced to phase III or represent a real treatment promise for anxiety disorders [103,104]. In the last period, the molecules mainly investigated for anxiety disorders have been psychedelics (often associated with psychotherapies), drugs with glutamatergic action, and cannabinoids. The development of new therapeutic agents has been impeded by methodological limitations in clinical trials, including the absence of appropriate control groups or the use of placebos rather than direct comparisons with established treatments such as SSRIs or benzodiazepines. This trend contrasts with the large number of ongoing studies for schizophrenia, mood disorders, and OCD.

We summarize the data available on the agents that were tested for treatment of anxiety disorders.

#### 4.4.1. Lysergic Acid Diethylamide

Lysergic Acid Diethylamide (LSD) is a well-known hallucinogen with psilocybin-like effects and is a Schedule I drug (into which are included drugs with no currently accepted medical use and a high potential for abuse, such as heroin, marijuana, etc.). It acts on serotonin receptors, with anxiolytic potential. To our knowledge, in the period under review, two trials about the use of LSD in subjects with anxiety disorders were published. These studies were one phase II study and one subsequent prospective study [105,106]. The studies included, respectively, 42 and 39 patients with anxiety disorders (PD, GAD, and social anxiety disorder) who received 200 µg of LSD or placebo for 2 sessions. The authors found that LSD produced lasting and noticeable reductions in symptoms of anxiety and comorbid depression for up to 16 weeks, with improvements in the following scales: State–Trait Anxiety Inventory—Global scores (STAI-G), STAI-State, STAI-Trait, Hamilton Depression Rating Scale-21 items version (HAM-D-21), Beck Depression Inventory (BDI), and SCL-90-R scores. The only severe side effect (transient acute anxiety episode) occurred in one patient [105]. In the prospective study, performed 12 months later, the LSD-assisted therapy for anxiety was demonstrated to have sustained long-term effects [106].

It is noteworthy that the FDA has recently designated MM-120 (lysergide D-tartrate) for general anxiety disorder (GAD) as a Breakthrough Therapy after a phase 2b study published in late 2024 (NCT05407064) but not currently available on PubMed (https://ir.mindmed.co/news-events/press-releases/detail/137/mindmed-receives-fda-breakthrough-therapy-designation-and-announces-positive-12-week-durability-data-from-phase-2b-study-of-mm120-for-generalized-anxiety-disorder, accessed on 24 April 2025).

#### 4.4.2. 3,4-Methylenedioxymethamphetamine

3,4-Methylenedioxymethamphetamine (MDMA) is a psychoactive compound traditionally classified as a psychedelic amphetamine that was considered a Schedule I controlled substance in the 1980s. In more recent years, MDMA obtained initial promising evidence in the treatment of post-traumatic stress disorder (PTSD). We found two recent studies on this drug: an RCT and its subsequent exploratory data analysis [107,108]. The duration of the RCT was 18 weeks. The trial was conducted on 90 patients with PTSD who were administered MDMA (80–180 mg) combined with 3 preparatory sessions and 9 sessions of integrative therapy. In the first study, MDMA was found efficacious at inducing a significant reduction in the Clinician-Administered PTSD Scale for DSM-5 (CAPS-5) score and of the total Sheehan Disability Scale (SDS) score compared with placebo. In the second trial [107], MDMA-assisted therapy facilitated a statistically significantly greater improvement, compared with placebo, on the Toronto Alexithymia Scale (TAS-20), the Self Compassion Scale (SCS), and on most factors of the Inventory of Altered Self-Capacities (IASC). These data indicate that, compared with inactive placebo therapy, MDMA-assisted therapy is highly effective in individuals with severe PTSD, and the treatment is safe and well tolerated, even in individuals with comorbidities (such as dissociation, depression, a history of alcohol and substance use disorders, and childhood trauma). MDMA did not induce adverse events of potential abuse, suicidality, or QT prolongation [108]. These data provide a possible window into understanding the psychological capabilities facilitated by psychedelic agents that may result in significant improvements in PTSD symptomatology [107].

MDMA-assisted therapy represents a potential innovative treatment that deserves rapid clinical evaluation. This intervention for PTSD has been granted Breakthrough Therapy designation by the FDA [98,108].

#### 4.4.3. D-Cycloserine

D-cycloserine (DCS) is an NMDA partial agonist, which we have already discussed in the section on DB. It is the most widely studied among glutamatergic agents in anxiety [109]. A 2015 Cochrane review reported no difference between DCS and placebo in increasing the effects of cognitive–behavioral therapy (CBT) in anxiety and related disorders in both children and adolescents [110]. A following meta-analysis found a small difference between DCS and placebo after treatment in patients with anxiety disorders but minimal gains in follow-up treatments [111]. Recent evidences, although not completely in agreement with each other, confirm these previous findings. Five studies were performed from 2018 onward on DCS: one on social anxiety disorder [112], two on panic disorder (PD) [113,114], and two on agoraphobia [115,116]. The first RCT included 169 participants with social anxiety disorders who received 50 mg of DCS plus CBT or placebo plus CBT during 12 sessions of exposure-based group CBT. Results showed that DCS did not potentiate the effects of CBT (no significant changes at the Liebowitz Social Anxiety Scale (LSAS)) [112]. The efficacy of DCS (50 mg) combined with CBT versus placebo was also tested in a subsequent RCT on 24 adolescents with PD. Both groups showed a reduction in anxiety symptoms with no significant differences between them [114]. The second RCT on PD obtained similar results: 33 PD patients were randomized to a single dose of DCS (250 mg) or placebo 2 h before an exposure therapy session. DCS was associated with greater clinical recovery at the 1-month follow-up than placebo, but the placebo group equaled the clinical gains of the DCS group at the 6-month follow-up [113]. The last 2 trials were conducted by Pyrkosch and collaborators [115,116] on 73 patients with agoraphobia with or without PD. These studies consisted of one RCT and one subsequent exploratory data analysis. In the double-blind RCT, individuals were treated with 12 CBT sessions comprising 3 exposures. After a successful exposure, patients received DCS (50 mg) or placebo. Authors observed that during CBT, patients’ symptomatology decreased significantly in both groups [116]. In the subsequent analysis, they investigated anxiety levels during the sessions and observed a decrease in several outcomes measured in subjective units of distress (SUDS) in the DCS group (i.e., initial anxiety, maximal anxiety during exposure, and within-session habituation), while the placebo group mostly maintained similar levels of anxiety from session to session. Thus, the results suggested potential short-term effects of DCS on anxiety during exposure, but not lasting benefits. Therefore, modifications may be desirable to reduce anxiety during exposure or to speed up the learning process while maintaining treatment efficacy. However, due to the exploratory nature of the analyses, confirmatory studies testing the hypothesis of a decrease in SUDS in agoraphobic patients are needed [115].

The use of DCS to augment exposure-based therapy for anxiety disorders showed mixed though initially positive effects that must be replicated in further trials.

#### 4.4.4. Ketamine

We have already discussed ketamine in the previous paragraphs. Originally developed as an anesthetic, ketamine has more recently demonstrated rapid and robust antidepressant effects in numerous studies. Several investigations have also explored its tolerability and potential efficacy in the treatment of anxiety disorders. An early study reported the beneficial effects of daily oral ketamine on both depressive and anxiety symptoms in adults in hospice care [117]. Another small open-label study suggested the benefit of ketamine administered subcutaneously in a single escalating dose design (up to 1 mg/kg) in patients with refractory social anxiety and/or GAD [118]. To our knowledge, five studies have been performed since 2018: an open-label study, three RCTs (placebo- or midazolam-controlled), and an exploratory double-blind psychoactive-controlled replication study [119,120,121,122,123]. In three of these studies, the administration was intravenous [120,122,123]; in the other two, it was subcutaneous [119,121]. The dosage of ketamine was variable, also depending on the patient’s weight. The patient samples ranged from 13 to 158 patients. An RCT in which intravenous ketamine was administered at 0.5 mg/kg compared with saline placebo showed a benefit in 18 patients with social anxiety measured using the Liebowitz Social Anxiety Scale (LSAS) [120]. Another study by Glue et al. [119] showed an early benefit of ketamine 1 mg/kg injected subcutaneously once or twice a week for 3 months among patients who had responded to the initial study with increasing doses. This result was confirmed in the subsequent and more robust study by Glue et al. [121]: they administered to 13 patients with an anxiety disorder ketamine at escalating doses (0.25, 0.5, 1 mg/kg) or midazolam (0.01 mg/kg) weekly (versus placebo) and found that ketamine may be a potential therapeutic option for patients with treatment-resistant GAD or social anxiety disorder. Furthermore, ketamine was found to be safe and well tolerated. Two RCTs specifically treated patients with PTSD [122,123]. In the randomized midazolam-controlled trial by Feder et al., 30 individuals with chronic PTSD received 6 infusions of ketamine (0.5 mg/kg) or midazolam (0.045 mg/kg) for 2 consecutive weeks. The ketamine group showed a significantly greater improvement in the total Clinician-Administered PTSD Scale for the DSM-5 (CAPS-5) and the MADRS scores than the midazolam group. Furthermore, 67% of participants in the ketamine group responded to treatment compared with 20% in the midazolam group. In contrast, the results of the study by Abdallah et al. were mixed. They administered 8 twice-weekly infusions of intravenous low-dose (0.2 mg/kg) or standard-dose (0.5 mg/kg) ketamine or placebo to 158 veterans and military personnel with PTSD. However, no significant differences were detected between the groups for PTSD symptoms as measured by PCL-5 or CAPS-5. As expected, however, the standard dose of ketamine improved depression, as measured by the MADRS, significantly more than placebo. In both RCTs, ketamine infusions were well tolerated, with no serious adverse events [122,123]. The main side effects were agitation [120,123], irritability, constipation [123], dissociative symptoms, blurred vision, drowsiness, nausea and vomiting [120,121], numbness, feeling of confusion/difficulty thinking, dizziness, distortion of the sense of time, tachycardia, hearing distortion, visual hallucinations, lightheadedness, feeling of floating, feeling of heaviness in the body, or changes in blood pressure [120]. Further studies are needed to understand the full potential of ketamine as a treatment for anxiety and, in particular, for chronic PTSD.

#### 4.4.5. Cannabinoids

There exists a prevalent misconception that cannabis, cannabidiol (CBD), and other cannabinoids are benign substances capable of alleviating anxiety and promoting relaxation. However, the current literature does not substantiate the notion that cannabinoids are safe for individuals with anxiety disorders, nor does it support the idea that cannabinoids effectively reduce anxiety or related symptoms in this patient population. The quality of evidence derived from clinical trials investigating cannabinoids for anxiety remains notably low. Endogenous and exogenous cannabinoids act on the cannabinoid receptor type 1 (CB1), the serotonergic receptor type 1A (5-HT1A), and the transient receptor potential vanilloid receptor type 1 (TRPV1). CB1 receptor agonists have a biphasic effect: they have anxiolytic properties at low doses and anxiogenic properties at high doses. While CB1 receptor activation produces an inhibitory impact on the neuron, leading to an anxiolytic effect, high doses of CB1 receptor agonists induce activation of the TRPV1 receptor, which produces anxiogenic effects. Several drugs that act as 5-HT1A receptor agonists have proven effective in the treatment of anxiety disorders, and recent studies indicate that cannabidiol and other cannabinoids also act on the 5-HT1A receptor, resulting in potential anxiolytic effects. The most studied cannabinoid in anxiety is CBD. Preclinical studies in animal models and human trials indicate that CBD is a potentially effective treatment for PD, GAD, and SAD [124].

We found four RCTs [125,126,127,128] and one open-label study [129] concerning the effects of CBD. The sample size ranged between 31 and 178 patients, whose diagnoses were agoraphobia [127], PTSD [128], or anxiety disorders [125,126,129]. The durations of the trials were between 8 days and 26 weeks. The CBD dosage was between 300 and 800 mg/day. An RCT showed that 300 mg/day for 4 weeks of CBD could be a useful option in the treatment of social anxiety in adolescents. Berger et al.’s open-label study suggests that CBD can reduce anxiety severity and has an adequate safety profile in young people with treatment-resistant anxiety disorders (CBT and/or antidepressant medication) [129]. However, this finding was not confirmed in a study, published in the same year, in adult PD patients with agoraphobia or social anxiety disorder [127]. Furthermore, a single 300 mg dose of CBD was shown to attenuate the increased anxiety and cognitive impairment induced by the re-enactment of a traumatic event in patients with PTSD when the event involved a non-sexual trauma, whereas when the traumatic event was of a sexual nature, no differences were seen [128]. In a very recent and encouraging study in 2024, nano-dispersible CBD was found to be therapeutically safe, free of serious adverse events, well tolerated, and effective for the treatment of mild and moderate anxiety disorders as well as associated depression and sleep quality disorders. These results pave the way for a probable prospective use of the nano-dispersible CBD formulation for various psychiatric disorders alone or in combination with other drugs [125].

The main adverse effects, which often occurred with both CBD and placebo [127], were fatigue, dizziness, bad mood, drowsiness, and hot flushes or chills [127,129]. No serious and/or unexpected adverse events occurred [129]. As is often the case when studying these molecules, the results are mixed. Controlled, randomized studies with different dosing regimens are therefore needed to confirm the long-term efficacy and safety of this compound.

The anxiolytic effects observed with CBD contrast with the anxiogenic effects induced by delta-9-tetrahydrocannabinol [130]. Delta-9-tetrahydrocannabinol (THC) is a CB1 receptor partial agonist that can have an anxiolytic effect at low doses, but at high doses, it can induce anxiety and panic attacks [131]. There are no registered studies for THC in anxiety disorders.

### 4.5. Obsessive-Compulsive Disorder

Once considered infrequent, obsessive-compulsive disorder (OCD) is now recognized as a critical source of psychological burden and public health concern [132,133]. OCD affects between 1% and 2% of the world’s population [134], with significant social and economic costs. Pharmacological and psychotherapeutic treatments for OCD are well established and supported by a strong evidence base [135]. SSRIs are the first-line treatment, useful for about half of all patients. Combination treatment with SSRIs and low-dose antipsychotics is an evidence-based second-line strategy. Cognitive–behavioral psychotherapy (CBT) is also used as both first- and second-line treatment and may be helpful for many patients. Unfortunately, a substantial minority of patients, perhaps up to one-third, derive little benefit from the available treatments. Therefore, there is an urgent need for new knowledge about the causes and pathophysiology of the disorder and new strategies for treatment, prevention, and cure [136].

Let us consider certain therapeutic strategies that are occasionally employed in treatment-refractory cases despite a limited evidence base. In the context of OCD, we cannot easily conclude, based on the established efficacy of SSRIs [137,138], that serotonin deficiency or dysregulation is central to the underlying pathophysiology of the disorder. Indeed, evidence from other sources provides limited support for this theory. Careful consideration of the drug-response pattern may give some clues as to what is going on in patients’ brains or at least facilitate the formulation of new hypotheses. It is widely recognized that OCD is extremely heterogeneous. This creates challenges both for pathophysiological studies and for the development and validation of new treatments. Genetic factors, which determine 40–50% of the risk, are complex and not well understood [139], and environmental or evolutionary contributions are even more unclear. A recent genetic analysis identified 2 candidate rare genetic causes for OCD and suggested the existence of as many as 400 similar large-effect rare mutations, each explaining a small fraction of cases [140]. Such rare causes may be decisive in individual cases but will never be observed in a population study [141,142], as they explain too small a percentage of the population variance. This heterogeneity complicates the search for new treatments. This characteristic makes it difficult to find statistically significant answers in large-scale studies. Clinical OCD may represent a common final pathway that individuals may reach through different trajectories [136]. Alternatively, it may not be a unitary condition at all, but rather a multidimensional spectrum [143] or a collection of phenomenologically overlapping pathophysiological entities that do not share a common core. We must be cautious about what we can expect from group-level drug treatment studies; given the heterogeneity of OCD, it is unlikely that any single treatment will produce relevant benefits in all patients. The last FDA approval of a drug for OCD was two decades ago. No drug we will discuss has been approved for the treatment of OCD by the FDA, the EMA, or other drug agencies.

#### 4.5.1. Pregabalin

Pregabalin is a gamma-aminobutyric acid analog authorized for epilepsy, neuropathic pain, fibromyalgia, and generalized anxiety disorder. The mechanism of action is not fully known: probably, it binds to voltage-dependent calcium channels with high affinity, reducing calcium currents and thus the release of several neurotransmitters, including glutamate [144]. Due to this presumed anti-glutamatergic effect, pregabalin has been tested anecdotally in patients with OCD. There is a 12-week, double-blind RCT on 56 patients with treatment-resistant OCD [145]. Pregabalin (mean dose of 185 mg) or placebo was added to sertraline therapy. A decline of over 35% in the Yale–Brown Obsessive-Compulsive Scale (Y-BOCS) score was registered in patients receiving pregabalin augmentation compared with placebo. Tolerability was good. Therefore, these results indicate that pregabalin can be considered a promising augmentation therapy.

#### 4.5.2. Memantine

Memantine is the glutamate modulator (NMDA receptor antagonist) most studied for the treatment of refractory OCD. Memantine is a low-affinity, open-channel NMDA receptor antagonist characterized by a rapid clearance rate. These pharmacokinetic and pharmacodynamic properties may account for its distinct clinical effects compared with those of high-affinity NMDA receptor antagonists such as ketamine, which will be discussed subsequently. Although mainly used to treat Alzheimer’s dementia, some placebo-controlled studies have suggested a benefit in treatment of OCD. We found two RCTs in which memantine was administered, either with ongoing SSRI-based therapy [146] or in addition to sertraline [147], to OCD patients, in samples of 32 and 70 individuals, for 12 weeks. Improvement was obtained in Y-BOCS total score, Y-BOCS obsession subscale score, Y-BOCS compulsion subscale score [146,147], and WCST number of categories subscale score [147]. The main side effects were headache, diarrhea, decreased appetite, insomnia, and vomiting [146,147]. Modarresi’s study (2018) and others were summarized in a systematic review suggesting a substantial benefit from memantine treatment in adults with OCD [148]. However, these placebo-controlled studies on memantine [146,149] are individually small and were performed in the same geographical area, Iran. They produced greater effects than those observed in open-label studies performed elsewhere, with response rates in individual studies as high as 100%, raising doubts about their generalizability. In light of these limitations, memantine cannot currently be considered as an established treatment for OCD. Further large-scale studies employing rigorous methodology and encompassing diverse geographical populations are warranted to more definitively evaluate its efficacy and safety [150].

#### 4.5.3. Tolcapone

Existing research on the use of dopamine antagonists as adjunctive treatments for OCD is considerably more robust than the limited evidence available on strategies designed to potentiate dopaminergic transmission. Several studies have evaluated the efficacy of catechol-O-methyltransferase (COMT) in disorders such as schizophrenia, DB, Parkinson’s disease, DDM, ADHD, addictions, and in OCD [151]. Along with evidence of benefit from psychostimulant treatments, there is one study on the pro-dopaminergic tolcapone, which further suggests that dopamine agonist approaches may offer relief from OCD symptoms. Tolcapone is a COMT inhibitor used in the treatment of Parkinson’s disease, and it enhances dopamine signaling in prefrontal cortical networks. It inhibits levodopa methylation by preventing the formation of its metabolite methyldopa [152]. Tolcapone was studied in OCD in a double-blind placebo-controlled crossover study including 20 patients. The authors found that tolcapone 200 mg/day was significantly superior to placebo in reducing obsessive-compulsive symptoms (Yale–Brown Obsessive-Compulsive Scale (Y-BOCS) score improvement) and was well tolerated. The main side effects were a reduction in bowel movements, fatigue, muscle aching, joint stiffness, and sleep problems. It should be noted that 25 percent of the participants were simultaneously receiving stable medication (SSRI or venlafaxine), and 80 percent had received treatment for OCD (SSRI or psychotherapy) in the past [153].

#### 4.5.4. Ondansetron

The serotonin 5-HT3 receptor antagonist ondansetron is used primarily as an antiemetic. Nevertheless, it has been also examined as augmentation treatment in OCD. Since serotonin 5-HT3 receptors are indirect inhibitors of dopamine release at the corticomesolimbic level, potentiation with the 5-HT3 receptor antagonist ondansetron in combination with SSRIs and antipsychotics has potential efficacy in patients with treatment-resistant obsessive-compulsive disorder [154]. Interestingly, ondansetron appears to reduce activation in cortical networks associated with interoceptive and sensorimotor processing, indicating a potential role in the management of OCD symptoms characterized by disgust reactivity and/or sensory over-responsivity [155]. To our knowledge, there are two RCTs on the use of this drug in OCD. The first [156] included 135 patients with treatment-resistant OCD and stable SSRI and antipsychotic therapy. Participants received ondansetron (4 mg), granisetron (2 mg), or placebo daily for 14 weeks. Granisetron is another serotonin 5-HT3 receptor antagonist. Y-BOCS score decreased in ondansetron, granisetron, and placebo groups by 41.5%, 39.7%, and 15.2%, respectively. Furthermore, the complete response in the ondansetron group was significantly higher than in the granisetron group. Relapse occurred in 3 (7.31%) patients who received granisetron, while it did not occur in patients who received ondansetron. The results of this study suggested the beneficial effects of ondansetron and, to a lesser extent, of granisetron as augmentation therapy in treatment-resistant OCD [156]. The findings of the second RCT are substantially in line with those of the previous study. The trial was conducted in a smaller sample (40 patients), in which ondansetron or placebo were administered, in addition to standard therapy, for 12 weeks. The authors found a significantly higher response rate at 8 and 12 weeks in patients treated with ondansetron compared with placebo. Ondansetron was generally well tolerated. Among adverse effects, it may contribute to QTc interval prolongation. So, it should be used with caution in patients at risk of arrhythmia or when combined with other drugs that may prolong the QTc interval [157]. The main side effect experienced with ondansetron therapy is diarrhea, while for granisetron it is constipation and headache [156,157].

#### 4.5.5. *N*-Acetylcysteine

*N*-acetylcysteine (NAC), an orally bioavailable prodrug of *L*-cysteine, promotes the synthesis of glutathione, an endogenous antioxidant capable of binding to and modulating glutamatergic neurotransmission via the AMPA and NMDA receptor systems. NAC was studied in patients with refractory OCD [158]. Results on the efficacy of this compound are not consistent [159,160,161]. Two recent RCTs were performed [162,163]. Li’s study was conducted on 11 boys aged 8–17 years and lasted 12 weeks. NAC was administered incrementally over the weeks up to 2700 mg/day in monotherapy. It showed an improvement in the Children Yale–Brown Obsessive-Compulsive Scale (CY-BOCS). The only side effect reported was a skin rash. On the contrary, Sarris’s study (2022) examined 89 OCD patients who received NAC as single therapy (500 mg/day) versus placebo for 20 weeks, but it did not find significant differences in Y-BOCS score between the two groups. Therefore, available evidence is still too scarce and uneven to draw conclusions.

#### 4.5.6. Methylphenidate

Methylphenidate, a central nervous system (CNS) stimulant, was largely studied for the treatment of attention deficit hyperactivity disorder (ADHD) and narcolepsy.

The idea that these stimulants might be useful in OCD seems counterintuitive. In fact, compared with ADHD, a condition well known to respond to stimulants, OCD shows the opposite pattern of frontostriatal activity (hyperactivation versus hypoactivation) and is at the opposite end of the compulsivity–impulsivity behavioral spectrum [164]. However, the neuropsychological literature has shown that the two conditions exhibit similar deficits in executive function, including response inhibition, attention allocation, and cognitive set shifting [164], possibly indicating a shared pathophysiology. Such deficits in OCD may underlie the beneficial effects of stimulant drugs. For instance, caffeine seems to attenuate experimentally induced distress and compulsive cleaning behavior in individuals with strong contamination fears. One hypothesis could be that caffeine-induced arousal has a beneficial effect on inhibitory control, and thus, on reducing compulsions [165].

Until now, only a single RCT evaluated the efficacy of methylphenidate in OCD patients. It consisted of a comparison between methylphenidate extended-release (MPH-ER) (until 36 mg/day) and placebo in 44 adults with OCD who were not sufficiently responsive to at least 8 weeks of fluvoxamine at the appropriate dosage [166]. A significant decrease in Y-BOCS scores was found in the MPH-ER + fluvoxamine group compared with the MPH-ER + placebo group. In particular, the Y-BOCS total score and Y-BOCS obsession subscale score were improved, together with anxious symptoms. No significant adverse effects were registered.

In conclusion, initial findings suggested that stimulants could represent a fast-acting pharmacotherapy augmentation option for treatment-resistant OCD patients. Long-term studies of larger samples are needed to assess both efficacy and abuse potential over time.

#### 4.5.7. Other Psychedelics

Over the years, several trials were performed to evaluate the suitability of use of psychedelics such as psilocybin, ketamine, and cannabidiol for the treatment of OCD [167,168,169,170,171,172]. Unfortunately, studies on psychedelic substances in OCD suffer from several methodological limitations. For example, some studies included healthy volunteers with sub-threshold OC symptoms or patients with other medical conditions; the rating scales are mainly self-administered; some studies are retroactive (with a risk of recall bias); protocols of study are not standardized; the dosages of drugs have not been specified. Furthermore, it is difficult to perform blind studies on psychedelic substances due to their psychoactive and hallucinogenic properties, even at very low doses [173]. We lack specific data on the effects of psychedelics on brain neurochemistry and connectivity in patients with OCD. Current animal models of OCD also have poor predictive validity, making it difficult to conclude the molecular mechanisms of psychedelics for compulsive behaviors. Legal restrictions on psychedelics in many countries make it difficult to assess the true extent of their use and benefits for psychiatric disorders.

All these considerations have made us desist from a systematic analysis of the available studies. In fact, the results cannot be considered sufficiently reliable.

## 5. Conclusions

In this review, we have identified some new drugs approved by the major drug agencies (FDA, EMA) for schizophrenia, BD, MDD, and anxiety disorders. Lumateperone and the combination OLZ/SAM are both atypical antipsychotics approved and tested for the treatment of schizophrenia and, mainly in association with mood stabilizers, for BD. OLZ/SAM reduces the adverse effects of olanzapine, such as weight gain, with the use of samidorphan. However, it does not induce a significant change in metabolic parameters, including insulin, glucose, glycated hemoglobin, and triglycerides. KarXT, recently approved for schizophrenia, is a drug combination in which the anticholinergic side effects of xanomeline are mitigated by trospium. Ketamine and esketamine represent promising therapeutic options for the treatment of treatment-resistant MDD and BD due to their rapid antidepressant and anti-suicidal effects. Ketamine has not yet been approved for any psychiatric disorders. In mood disorders, it has shown significant benefits in patients with mixed features and high suicide risk, although its use remains off-label. Ketamine has also been studied in anxiety disorders and PTSD, with promising but still inconclusive results. Esketamine, administered by nasal spray, has been approved for the treatment of TRD and acute suicidal ideation. However, some limitations persist, including the short duration of the antidepressant effects, the need for optimized dosing strategies, and uncertainties about long-term safety. Further research is crucial to establish effective and sustainable treatment protocols for patients with TRD and bipolar depression. The combination of dextromethorphan and bupropion has been approved for the treatment of MDD. It represents an encouraging option due to its novel mechanism of action, which includes modulation of NMDA receptors and inhibition of dopamine and noradrenaline reuptake. Studies showed good efficacy and tolerability, although further long-term follow-up investigations are needed.

Brexanolone, the first approved treatment for PPD, provides rapid and substantial effects. However, its intravenous administration and high cost limit its accessibility. Future advances, such as oral formulations or alternative treatments with similar mechanisms of action, could broaden its clinical application. Zuranolone has recently been approved for PPD, offering a more convenient administration regimen than brexanolone (oral vs. intravenous administration). Both drugs represent an important advance but require further research to optimize their clinical applications and accessibility. In addition, the combination of DCS and lurasidone (NRX-101) showed superiority over lurasidone only in maintaining improvements in depressive symptoms and reducing suicidal ideation. DCS has been extensively studied in anxiety disorders and particularly in PD, suggesting positive initial effects, which need to be replicated in further studies.

Anti-inflammatory agents, such as celecoxib and infliximab, are emerging as additional options for the treatment of bipolar depression (in particular, in preventing relapses), although the evidence is still evolving. The results emphasize the importance of a personalized approach in the exploration and application of anti-inflammatory therapies to identify the patients most likely to obtain benefits.

Ulotaront, a TAAR1 agonist with 5-HT1A agonist activity, is a useful compound in patients with psychosis also due to its lack of extrapyramidal symptoms or metabolic side effects. It has been designated a Breakthrough Therapy by the FDA. Another antipsychotic under consideration is ruloperidone, which acts by blocking serotonin, δ-, and α-adrenergic receptors and is targeted at negative symptoms in patients with schizophrenia.

The use of hallucinogens has been making a comeback in recent years: LSD, MDMA, and psilocybin have been tested in anxiety disorders. LSD and MDMA were found efficacious in reducing the severity of anxious symptoms, mainly in combination with psychotherapy, and were designated as Breakthrough Therapies by the FDA, respectively, LSD for GAD and MDMA for PTSD. Psilocybin showed prolonged antidepressant effects in TRD, albeit in preliminary studies. Despite its therapeutic potential, important questions remain about its regulation, safety, and clinical applicability. Future studies should clarify its role in relation to standard treatments and establish guidelines for its safe and effective use. Studies on cannabidiol have produced mixed results. It seems to be useful in treating anxiety disorders in young people. Nevertheless, the potential for abuse must not be overlooked, and therefore, selected cases must be chosen and use must be monitored very carefully.

Pregabalin, memantine, tolcapone, ondansetron, NAC, and methylphenidate seem to produce some benefits, with varying degrees of evidence, in treating OCD. Overall, studies are too limited to draw any final conclusion.

In conclusion, the available literature on the new advances in pharmacotherapy for psychiatric disorders suggests that the direction of research is to identify new molecules or new formulations of existing drugs in order to personalize treatments as much as possible while minimizing adverse effects. New research projects and increasing funding to collect more complete data on psychiatric treatments will be needed to reduce the burden of these diseases and improve patients’ quality of life.

As medical research increasingly moves in the direction of precision medicine and tailored treatments, psychiatric research should take advantage of studies considering both pharmacogenetic and pharmacogenomic approaches in order to detect genetic variations associated with treatment outcomes in psychiatry, including the response, resistance, and adverse effects rate of medications. Moreover, the approach of digital phenotyping (a new strategy that uses measures extracted from spontaneous interactions with smartphones to track mental status) should be implemented in order to monitor long-term effects of medications in geographical areas where access to mental health services is more difficult.

## Figures and Tables

**Figure 1 pharmaceuticals-18-00665-f001:**
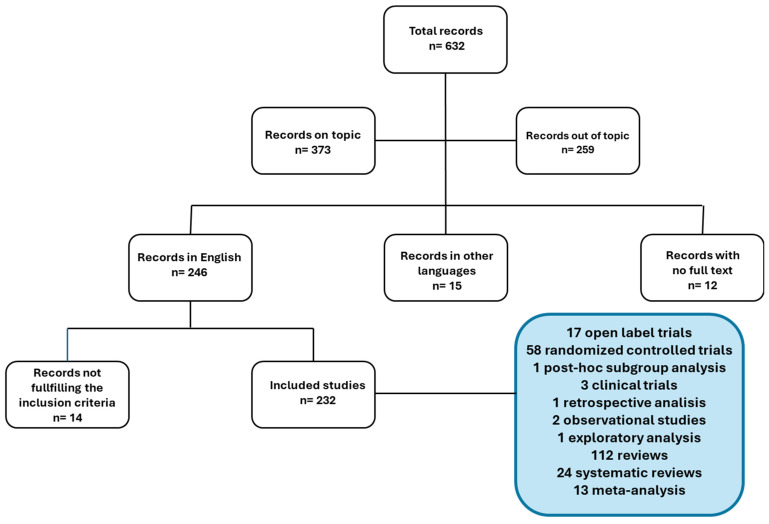
Literature search flowchart.

**Table 1 pharmaceuticals-18-00665-t001:** Schizophrenia.

Drug	Approval Date	Diagnosis	Study Design	Dosage	Cohort	Treatment Duration	Results	Adverse Effects
Lumateperone [6]	2019	Schizophrenia, bipolar depression	Randomized double-blind placebo-controlled trial	60 mg/day, 120 mg/day lumateperone; or 4 mg/day risperidone	334	4 weeks	Improvement in PANSS score and depressive symptoms (in patients with negative symptoms at baseline)	Dry mouth, worsening symptoms of schizophrenia
Lumateperone [7]	2019	Schizophrenia, bipolar depression	Randomized double-blind placebo-controlled trial	28 mg/day or 42 mg/day	450	4 weeks	Improvement in PANSS, PANSS positive subscale, CGI-S, CGI-I	Sedation, fatigue, constipation, orthostatic hypotension, convulsion
Lumateperone [8]	2019	Schizophrenia, bipolar depression	Open-label study	42 mg/day	301	6 weeks	No significant change in PANSS score between baseline and the end of the study	Somnolence, headache, dry mouth
Lumateperone [9]	2019	Schizophrenia, bipolar depression	Pooled post hoc analysis (tolerability study)	Dosage of [6,7,8]	1073	4 weeks 4 weeks 6 weeks	Adverse events with lumateperone were equal to placebo and less than risperidone	Somnolence/sedation, dry mouth
RBP-7000 (risperidone subcutaneous) [10]	2018	Schizophrenia	Phase III open-label study	90 mg or 120 mg	500	52 weeks	No significant change in PANSS score and CGI score between baseline and the end of the study	73.4% reported at least one adverse event. Akathisia, tremor, extrapyramidal disorder, muscle spasms
RBP-7000 (risperidone subcutaneous) [11]	2018	Schizophrenia	Phase III single-arm open-label study	120 mg	482	52 weeks	Health-related quality of life and the subjective well-being under neuroleptic treatment remained stable. Increase in treatment satisfaction and preference of treatment at the end of the study	Not available
TV46000 (risperidone subcutaneous) [12]	2023	Schizophrenia	Clinical trial	50–125 mg q1m or 100–250 mg q2m	543	90 total relapse events occurred	Improvement in social functioning and quality of life	Injection site nodules, ↑ weight, extrapyramidal symptoms
TV46000 (risperidone subcutaneous) [13]	2023	Schizophrenia	Randomized double-blind controlled trial	50–125 mg q1m or 100–250 mg q2m	331	56 weeks	Improvement in quality of life	Worsening symptoms of schizophrenia, injection site pain, injection site nodule
Risperidone LAI (IM, administered every 28 days) [7]	2022	Schizophrenia	Randomized, placebo-controlled trial	75 or 100 mg	437	12 weeks	Improvement in PANSS, CGI-S, CGI-I	Hyperprolactinemia, akathisia, headache
Risperidone LAI (IM, administered every 28 days) [14]	2022	Schizophrenia	Open-label extension of PRISMA 3 study	75 or 100 mg	205	52 weeks	Improvement in PANSS total score, PANSS positive and negative subscale score, CGI-S, CGI-I	Headache, hyperprolactinemia, asthenia, ↑ weight, insomnia, akathisia
Transdermal asenapine [15]	2019	Schizophrenia	Phase III, randomized, placebo-controlled trial	7.6 mg/24 h, 3.8 mg/24 h	607	6 weeks	Improvement in PANSS. The systemic safety profile is like that of sublingual asenapine.	Erythema in the application site, headache, extrapyramidal disorder, insomnia, ↑ weight, anxiety, constipation, agitation, worsening symptoms of schizophrenia
OLZ/SAM [16]	2021	Schizophrenia, BD	Randomized, double-blind, placebo-controlled proof of concept study	10 mg/day OLZ, 10 mg/day OLZ + 5 mg/day SAM, 5 mg/day SAM	106	3 weeks	↓ weight for OLZ/SAM was less than compared with OLZ	Orthostatic hypotension, somnolence, nausea, abnormal liver function test
OLZ/SAM [17]	2021	Schizophrenia, BD	Randomized, double-blind phase II placebo-controlled study	OLZ plus placebo or OLZ plus 5–10 mg/day, or 20 mg/day of SAM	347	12 weeks	37% lower weight gain with OLZ/SAM5–20/5 mg/day compared with OLZ/placebo	Somnolence, sedation, dizziness, constipation
OLZ/SAM [18]	2021	Schizophrenia, BD	Phase I single-center, open-label, randomized study	10 mg/day OLZ/10 mg/day SAM, 10 mg/day OLZ	48	Not available	OLZ/SAM does not affect PK and bioavailability of OLZ	Dizziness, somnolence, nausea, tachycardia, sedation, headache, ↑ CPK, syncope
OLZ/SAM [19]	2021	Schizophrenia, BD	Two multicenter, open label, parallel-cohort studies (tolerability study)	A single oral dose of 5 mg/day OLZ plus 10 mg/day SAM	41	6 weeks	Well tolerated in hepatic and renal impairment	Somnolence, dizziness, nausea, abdominal pain, lethargy
OLZ/SAM [20]	2021	Schizophrenia, BD	Phase III, double-blind, randomized, active-(olanzapine) and placebo-controlled study	20/10 mg/day OLZ/SAM, 20 mg/day OLZ	401	4 weeks	Significant improvements in PANSS, CGI-S, and CGI-I with both OLZ/SAM and OLZ	↓ weight, somnolence, dry mouth, anxiety, headache
OLZ/SAM [21]	2021	Schizophrenia, BD	Phase III multicenter, randomized, double-blind study	10 mg/day OLZ/10 mg/day SAM and 20 mg/day OLZ/10 mg/day SAM	561	24 weeks	↓ weight	Somnolence, dry mouth, ↑ appetite
OLZ/SAM [22]	2021	Schizophrenia, BD	Open-label extension study	10/10, 15/10, or 20/10 mg/day OLZ/SAM	100	52 weeks	No significant QTc effect	Somnolence, ↑ weight, nausea, dizziness, constipation
OLZ/SAM [23]	2021	Schizophrenia, BD	Phase I, randomized, double-blind, placebo- and positive (moxifloxacin)-controlled, parallel-group, multiple-dose study	Subjects in the active arm received a therapeutic dose of 10/10 mg/day OLZ/SAM on days 2–4, 20/20 mg/day on days 5–8, and a supratherapeutic dose of 30/30 mg/day	42	2 weeks	Steady-state for OLZ took 3–4 days and SAM took 5 days. Different levels of OLZ had no impact on the pharmacokinetic profile of SAM.	Somnolence, ↑ weight, nausea, dizziness
OLZ/SAM [24]	2021	Schizophrenia, BD	Open-label extension study [21]	10 mg/day/10 mg/day OLZ/SAM, 15 mg/day/10 mg/day OLZ/SAM, or 20/10 mg/day OLZ/SAM	265	52 weeks	Weight, waist circumference, metabolic parameters, and symptoms of schizophrenia remained stable with OLZ/SAM even after 1 year	↑ glycosylated hemoglobin and psychotic disorder, headache
Xanomeline/Trospium (KarXT) [25]	2024	Schizophrenia	Phase II, randomized, double-blind, placebo-controlled (EMERGENT 1)	Starting with 50 mg/day of xanomeline and 20 mg/day of trospium twice daily and increased to a maximum of 125 mg/day of xanomeline and 30 mg/day of trospium twice daily, with the option to return to 100 mg/day of xanomeline and 20 mg/day of trospium twice daily	182	5 weeks	Improvement in PANSS, PANSS positive symptoms subscore, PANSS negative symptoms subscore	Constipation, nausea, dyspepsia, headache, somnolence, akathisia, dizziness, ↑ weight, tachycardia, diarrhea, ↑ γGT level, agitation, insomnia, ↓ appetite, hyperhidrosis
KarXT [26]	2024	Schizophrenia	Post hoc analyses of safety and tolerability [25] (EMERGENT 2)	Dosages were the same as Brannan’s study (2021) [25]	179	5 weeks	Reduction in the procholinergic and anticholinergic effects	Somnolence, sedation
KarXT [27]	2024	Schizophrenia	Phase III randomized, double-blind, placebo-controlled, flexible-dose study (EMERGENT 3)	Xanomeline/trospium was started with 50 mg/day xanomeline and 20 mg/day trospium twice a day, and then increased to 100/20 mg/day	256	5 weeks	Improvements in PANSS total score, PANSS negative factor score, CGI-S	Constipation, dyspepsia, headache, nausea, hypertension, dizziness, gastroesophageal reflux disease, diarrhea

Abbreviations: PANSS = Positive and Negative Syndrome Scale, ↑ = increase, ↓ = reduction, CGI-S = Clinical Global Impression—Severity Scale, CGI-I = Clinical Global Impression on Improvement, OLZ/SAM = olanzapine/samidorphan, PK = pharmacokinetic profile, CPK = Creatine Phosphokinase, LAI = long-acting injectable formulation.

**Table 2 pharmaceuticals-18-00665-t002:** Bipolar disorder.

Drug	Approval Date	Diagnosis	Study Design	Dosage	Cohort	Treatment Duration	Results	Adverse Effects
Lumateperone [28]	2021	Schizophrenia, bipolar depression	Phase III randomized placebo-controlled trial	42 mg/day or placebo	377	6 weeks	Improvement in depressive symptoms (↓ MADRS and the CDSS) (in patients with negative symptoms at baseline)	Somnolence, nausea
Lumateperone [29]	2021	Schizophrenia, bipolar depression	Randomized, placebo-controlled trial (different dosages)	28 mg/day, 42 mg/day	528	6 weeks	Improvement in overall symptoms severity and a reduction in the functional impact of symptoms. (↓ CGI-BP-S and SDS scores)	Somnolence, dizziness, nausea
OLZ/SAM [30]	2021	Schizophrenia, BD	Phase III randomized controlled trial	OLZ/SAM 10–20 mg/day OLZ + 10 mg/day SAM, vs. OLZ 10–20 mg/day	426	12 weeks	↓ weight and waist changes in OLZ/SAM group vs. OLZ group; comparable reduction in disease severity in the two groups assessed by CGI-S (OLZ vs. OLZ/SAM)	↑ weight mainly in OLZ group, somnolence

Abbreviations: MADRS = Montgomery–Åsberg Depression Rating Scale, CGI-S = Clinical Global Impression—Severity Scale, CGI-BP-S = Clinical Global Impression—Bipolar Severity Scale, SDS = Sheehan Disability Scale, CDSS = Calgary Depression Scale for Schizophrenia, ↑ = increase, ↓ = decrease, BD = bipolar disorder.

**Table 3 pharmaceuticals-18-00665-t003:** Major depressive disorder.

Drug	Approval Date	Diagnosis	Study Design	Dosage	Cohort	Treatment Duration	Results	Adverse Effects
Esketamine [31]	2019	TRD and acute SI	Phase III, open-label trial	Esketamine 28 mg (starting dose age ≥ 65 years), or 56 mg, or 84 mg	1148	4 weeks of drug administration twice a week (average 31.5 months maintenance phase)	Improvement in overall symptom severity and a reduction in the functional impact of symptoms (↓ MADRS, ↓ CGI-S, ↓ SDS, ↓ PHQ-9)	Dissociation, dizziness, nausea, headache, ↑ blood pressure, sedation, urinary tract infections, ↑ liver enzymes, suicidal ideation, mania/hypomania
Dextromethorphan/bupropion (AXS-05) [32]	2022	MDD	Randomized, double-blind, multicenter, parallel-group trial	45 mg/105 mg/day dextromethorphan/bupropion or 105 mg/day bupropion for the first 3 days and twice daily thereafter	80	6 weeks	Improvement in depressive symptoms. Clinical response rates at week 6 in 60.5% in the dextromethorphan/bupropion group vs. 40.5% in bupropion group	Dizziness, nausea, dry mouth, decreased appetite, and anxiety
Dextromethorphan/bupropion (AXS-05) (GEMINI) [33]	2022	MDD	Phase III randomized, controlled trial	45 mg/day–105 mg/day	327	6 weeks	Improvement in depressive symptoms and reduction in CGI scores. Clinical response of 54% at 6 weeks.	Dizziness, nausea, headache, drowsiness, dry mouth
Brexanolone [34]	2019	PPD	Phase III randomized, controlled trial	During hours 0–4, 30 μg/kg/h; during hours 4–24, 60 μg/kg/h; during hours 24–52, 90 μg/kg/h; during hours 52–56, 60 μg/kg/h; during hours 56–60, 30 μg/kg/h	21	30 days	Improvement in depressive symptoms (↓ HAM-D total score)	No SAE, dizziness, somnolence
Brexanolone—Study 1 [35]	2019	PPD	Phase III randomized, placebo-controlled trial	60 μg/kg/h, 90 μg/kg/h	138	30 days	Improvement in depressive symptoms (↓ HAM-D total score)	Dizziness, somnolence, 1 SAE (suicidal ideation and intentional overdose attempt during follow-up; altered state of consciousness and syncope)
Brexanolone—Study 2 [35]	2019	PPD	Phase III randomized, placebo-controlled trial	90 μg/kg/h	108	30 days	Improvement in depressive symptoms (↓ HAM-D total score)	Dizziness, somnolence, 1 SAE (suicidal ideation and intentional overdose attempt during follow-up; altered state of consciousness and syncope)
Zuranolone (SAGE-217) [36]	2021	PPD	Phase III double-blind trial, placebo-controlled	30 mg/day	153	45 days	Improvement in depressive symptoms (↓ HAMD)	Drowsiness, dizziness, sedation
Zuranolone (SAGE-217) [37]	2023	PPD	Phase III double-blind trial, placebo-controlled	50 mg/day	196	45 days	Improvement in overall symptoms (↓ PHQ-9, ↓ HAM-D, ↓ HAM-A, and ↓ CGI-S scores)	Headache, diarrhea

Abbreviations: MADRS = Montgomery–Åsberg Depression Rating Scale, CGI-S = Clinical Global Impression—Severity Scale, PHQ-9 = Patient Health Questionnaire 9 item, SDS = Sheehan Disability Scale, HAM-D = Hamilton Depression Rating Scale, HAM-A = Hamilton Anxiety Rating Scale, SI = suicidal ideation, TRD = treatment-resistant depression, MDD = major depressive disorder, PPD = postpartum depression, SAE = serious adverse event, ↓ = decrease, ↑ = increase.

## Data Availability

Not applicable.

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
