# Peer review of "New Agents in the Treatment of Psychiatric Disorders: What Innovations and in What Areas of Psychopathology?"

_pharmaceuticals, 2025, doi:10.3390/ph18050665_

Round 1

Reviewer 1 Report

Comments and Suggestions for Authors

The authors did not include the keywords. Keywords are essential for indexing and facilitating searches.

Introduction:

-The introduction provides relevant and updated information regarding the global burden of psychiatric disorders, with a well-placed emphasis on treatment resistance and disease-related disability (e.g., DALYs). However, several points could be improved to enhance clarity, academic tone, and overall cohesion:

Some sentences require refinement for improved clarity and grammatical accuracy. For instance:

“proceeds to a lower pace” → should be “proceeds at a slower pace”

“new mode of administration” → should be “new modes of administration”

- Portions of the introduction repeat content from the abstract (e.g., statistics on DALYs and treatment resistance). Consider rephrasing or condensing these to avoid duplication.

- The rationale for selecting the period 2018–2025 is not explained. It would be helpful to briefly clarify whether it reflects a significant shift in regulatory trends, increased approvals, or another relevant factor.

- Does the review include inclusion/exclusion criteria? While this is a narrative review, it would be beneficial to clarify whether there were any minimal criteria for inclusion of agents (e.g., clinical trial phase, peer-reviewed publication, preliminary evidence threshold).

- The definition of DALYs provided could be more concise and technical, or moved to a footnote. The target readership likely has familiarity with this concept.

Overall, the introduction is relevant and sets the stage well for the review, but it would benefit from minor language improvements and further contextual clarification.

Methods:

- The search string is extensive but highly convoluted and may compromise reproducibility. It is recommended to: (1) Move the full search string to a supplementary appendix or online resource. (2) Summarize the search strategy in-text (e.g., “We searched PubMed using a combination of MeSH terms and free-text keywords related to psychotropic agents and psychiatric diagnoses.”). (3) Consider using a PRISMA diagram or flowchart to show the screening process, especially since inclusion/exclusion involved multiple steps.

- There is contradiction in the type of studies included vs. excluded. It states that observational studies were both included (line 109) and excluded (line 111). Likewise, open-label studies are listed among both included and excluded designs depending on phrasing.

Please clarify and list explicitly which study designs were accepted (RCTs, open-label trials, meta-analyses, etc.), and which were not. A table or bullet point format may help.

-  No information is given about (1) quality assessment or risk of bias of the included studies (Was any tool used? e.g., GRADE, Cochrane RoB?); (2) number of reviewers involved in screening and how disagreements were handled. (3) whether non-indexed trials or grey literature were considered (e.g., ClinicalTrials.gov, EMA/FDA databases directly).

- The tables themselves are rich in content but need: (1) standardization of terminology and abbreviations (e.g., consistently write "Randomized Controlled Trial", not "RCT" in one row and full in another); (2) Clearer column headers (e.g., “Results” column sometimes includes both efficacy and tolerability); (3) You may want to add a separate “Adverse Events” column for consistency; (4) Use of abbreviation keys is helpful—ensure they are consistent across tables and not duplicated (e.g., PANSS, CGI-S); (5) It would be clearer if drugs were grouped by indication or by regulatory approval year in the tables.

4.1. Schizophrenia

- Several receptor mechanisms (e.g., D2, 5-HT1A) are repeated across the text. Consolidating these explanations would streamline the section and improve flow.

- Some sentences are overly complex or lengthy, which may hinder reader engagement. Consider breaking them into shorter, more digestible parts.

- When listing newer agents (e.g., lumateperone, ulotaront, ruloperidone), it would be helpful to clearly distinguish between those already approved and those still under investigation.

- The mention of LAIs is a strength. This section could be enriched by specifying which LAI formulations are currently available or most effective, as this has major clinical implications.

4.1.1. Lumateperone

- While it is important to mention that Lumateperone was FDA-approved in 2019, it would be useful to briefly explain what therapeutic alternatives were available at the time for the treatment of schizophrenia and depressive episodes in bipolar disorder. This would help situate Lumateperone within the clinical context.

- The pharmacological profile of Lumateperone is well described, but a slightly more detailed explanation of how the effects on the 5-HT2A, D1, and D2 receptors contribute specifically to therapeutic effects (e.g., reduction of positive and negative symptoms of schizophrenia) would enhance understanding of the drug's distinct action compared to other antipsychotics.

- While the description of the clinical trials is thorough, it would be helpful to mention the total duration of the studies (e.g., whether the 4 weeks mentioned is the total treatment duration in both trials). Additionally, it could be useful to provide more detail about the sample characteristics, such as age range and comorbidities, to give a more complete picture of clinical applicability.

4.1.2. Risperidone Long-Acting Injectable

-The section starts by clearly presenting the various forms of risperidone administration, but some information about the different formulations may seem redundant. For example, the repetition of details about intramuscular and subcutaneous administration could be condensed more efficiently, especially if the focus is on the benefits and study results.

- The sentence "one of the main advantages of subcutaneous injection is that it avoids damage to muscle tissue" could be better contextualized, explaining why this is relevant compared to the previous intramuscular formulations.

-The description of the clinical trials is good, but it could be more focused. Many details about dosages and study types are provided, but it would be helpful to better summarize the main findings of the trials in a more concise manner, perhaps first highlighting the most significant results (e.g., efficacy, safety, tolerability).

- Including data on adverse events is important, but it could be better structured to facilitate understanding. Maybe a table or list of the most frequent adverse events throughout the study could improve readability.

4.1.3. Transdermal Asenapine

- It is important to mention more clearly the purpose of the transdermal formulation in comparison with the sublingual one, emphasizing the advantage of being unaffected by food or drink intake. The transdermal delivery can be highlighted as a relevant factor in treatment adherence.

- Although the availability of three patch dosages (3.8, 5.7, and 7.6 mg/24 h) is mentioned, a more detailed explanation of how these doses were chosen and how they relate to the sublingual dosage would be interesting. Additionally, comparing the effects of each of the mentioned doses could clarify if the treatment response varies significantly with the dose.

4.1.4. Olanzapine/Samidorphan

- It's important to highlight the comparison between OLZ/SAM and monotherapy OLZ, as well as the efficacy of the clinical studies. The explanation of how the combination improves the metabolic profile, especially regarding weight gain, is relevant, as this is a significant adverse effect associated with Olanzapine.

-Mentioning the most common adverse events (drowsiness, dry mouth, and suicidal ideation) is essential, but it would be helpful to have a more explicit reference to the frequency of these adverse events in percentage terms, as done in other sections.

-The text could be enriched by providing more information about the benefits and challenges of using OLZ/SAM compared to previous treatments or other pharmacological combinations.

- The authors could mention that clozapine, developed about 70 years ago, represents a milestone in the development of antipsychotics. While it is particularly effective in treatment-resistant cases, its use is limited due to severe side effects, such as agranulocytosis, requiring regular monitoring.

4.2. Bipolar Disorders

- Lumateperone and samidorfan are mentioned with little explanation, especially regarding their impact compared to more traditional treatments. It would be interesting to expand on how these drugs work in the treatment of bipolar disorder.

- The text mentions that the dopaminergic, serotonergic, GABAergic, and glutamatergic systems are key targets in pharmacological treatment. A more detailed explanation of how each of these systems affects bipolar disorder could enrich this section.

- Why do approximately one-third of patients with bipolar disorder not respond to at least two treatment options and are considered treatment-resistant?

- What characterizes a mixed episode in bipolar disorder, and why is it associated with a higher risk of suicide?

- What are the most common complications associated with bipolar disorder besides the risk of suicide? How can treatment minimize these complications?

4.2.1. Lumateperone

- The text clearly describes the comparison with placebo, but it would be interesting to know if there is a comparison with other effective treatments for bipolar depression, aside from placebo, to better contextualize the efficacy of lumateperone.

4.2.2. Olanzapine/Samidorphan

- Did OLZ/SAM show different or more pronounced adverse effects in other areas (such as glucose metabolism, lipids, or sedation) compared to olanzapine?

- What is the comparative efficacy of OLZ/SAM in terms of controlling manic or psychotic symptoms, in addition to weight reduction?

4.2.3. Ketamine and Esketamine

- How do the effects of ketamine and esketamine vary in patients with specific characteristics of bipolar disorder, such as those with mixed episodes or psychotic symptoms?

- Is the efficacy of ketamine and esketamine comparable between patients with bipolar disorder type I and type II?

- How do ketamine and esketamine specifically affect anxiety, irritability, and agitation (AIA) symptoms in patients with BD, and how does this compare to their effects on depressive symptoms?

- What are the rates of remission and relapse for patients treated with ketamine or esketamine over the long term?

4.2.5. Celecoxib

- The text mentions that CBX has been investigated as an adjunct to lithium and valproate. However, it may be beneficial to further explain the rationale behind using celecoxib, given its COX-2 inhibition and anti-inflammatory properties. A detailed explanation of how inflammation may contribute to BD and how CBX could influence the mood stabilization process would strengthen the argument for its use.

4.3. Major Depressive Disorder

- The section mentions promising new treatments for MDD, including s-ketamine and dextromethorphan-bupropion, which have garnered attention for their potential efficacy in treatment-resistant depression (TRD). However, more detailed information about the mechanisms of action of these new treatments, as well as their comparative efficacy and safety relative to traditional treatments (e.g., SSRIs and SNRIs), would strengthen the section.

4.3.2. Dextromethorphan-Bupropion

-How does the combination of dextromethorphan and bupropion compare in terms of effectiveness and safety to other novel antidepressant treatments like esketamine or brexanolone?

- Given the relatively short duration of the trials (up to 15 months), what monitoring strategies should be in place for patients receiving long-term dextromethorphan-bupropion treatment to ensure sustained benefit and early detection of any adverse effects?

4.3.4. Zuranolone

- What is the long-term impact of zuranolone on patients with chronic depression or treatment-resistant depression (TRD), and how does it compare to other treatments like SSRIs, SNRIs, or brexanolone in such populations?

- What additional research is needed to explore the broader indications for zuranolone and its long-term safety, particularly in populations such as pregnant women, the elderly, or patients with comorbid conditions?

4.4. Anxiety Disorders

- How can clinical trials for anxiety disorders be improved to increase the chances of discovering effective new treatments?

- How can we better address the issue of treatment adherence in anxiety disorders and reduce the side effects of current first-line treatments?

4.5.4. Ondansetron

- What are the advantages of using ondansetron as an augmentation therapy in patients with treatment-resistant OCD?

-How do the effects of ondansetron compare to other augmentation treatments for OCD, such as clomipramine or other antipsychotics?

-What are the potential side effects of other 5-HT3 antagonists, such as granisetron, in the treatment of OCD?

-What clinical precautions should doctors take when prescribing ondansetron to patients with treatment-resistant OCD?

4.5.5. N-Acetylcysteine

- The "N" in "N-Acetylcysteine" should be italicized

-

References:

- I would like to point out that the authors should use square brackets [ ] instead of parentheses ( ) when citing sources in the text, for example: [157], [159], [160], as per the journal's guidelines.

Author Response

Answers to reviewer1

Thank you for your suggestions.

We have added the keywords.

Introduction:

-Thanks for your suggestions, we have clarified some sentences in the text.

- We feel that repetitions between the abstract and the rest of the text are necessary, as the abstract is a summary of the work done.

-We have clarified the inclusion and exclusion criteria.

Methods:

- We believe that reporting the entire search string in the text is useful to ensure the reproducibility of the work. We have already included a diagram explaining the screening process of the studies.

- Thank you for noticing the error, we have better explained which types of studies were included and which were excluded; we have included the required bulleted list.

-As the present is a narrative review, we did not use tools such as GRADE, Cochrane RoB. We have only once considered data not found in PubMed and it is made explicit as requested.

-Tables: we double-checked the standardisation of terminology and abbreviations and found no inconsistencies, we used ‘results’ as a column heading precisely to group all possible outcomes, the ‘adverse effects’ column already exists. The choice of creating three separate tables according to pathology and then sorting the drugs by the same logic as in the text is intended to make them easier to consult.

4.1. Schizophrenia

- Thank you for your comments, throughout the text we have briefly mentioned the mechanisms of action, focusing on the aspects we considered most relevant according to the type of drug and the symptomatology under consideration.

-We have revised the text to make the sentences more understandable.

- We highlighted better which drugs were approved and which were not, both in the ‘methods’ and in the ‘discussion’.

-There are currently many drugs with LAI formulations, we have not expanded on the introductory part of the text on schizophrenia in order not to make it heavy to read, but in the paragraphs explaining the new LAI drugs, we have made the existing formulations explicit.

4.1.1. Lumateperone

-As previously explained, we have not dwelled on long-established clinical and pharmacological aspects, precisely because the focus of this manuscript is on pharmacological innovations.

-We have included more details of the lumateperone trials section as requested.

4.1.2. Risperidone Long-Acting Injectable

- we edited the section on subcutaneous risperidone taking into account the points you have highlighted to us.

- The tables are intended to represent what you requested, including the treatment of adverse events.

4.1.3. Transdermal Asenapine

The correspondence between transdermal and sublingual dosage had already been made explicit, as well as the advantage of not being influenced by food or drink intake. The other measures requested have been carried out.

4.1.4. Olanzapine/Samidorphan

-Thank you for your comment, we have better explained the mechanism of action of samidorfan and explained the side effects of OLZ/SAM therapy.

-Clozapine is not discussed in our paper, although it is an efficacious drug in the treatment of resistant schizophrenia, because it has been used for many years and would therefore be off-topic. The topic of this review is the recent advances in pharmacotherapy of psychiatric disorders.

4.2. Bipolar Disorders

As this is a manuscript for a pharmacologically oriented journal, we have deliberately avoided going into lengthy descriptions of the clinical aspects of the disorders.

4.2.1. Lumateperone

We agree with your point, but we have not found any studies comparing lumateperone with other drugs.

4.2.2. Olanzapine/Samidorphan

The comparative efficacy of OLZ/SAM in the control of manic or psychotic symptoms is described in the table.

4.2.3. Ketamine and Esketamine

We have not dealt with certain aspects that you have emphasised because there are no significant data in the literature, furthermore, the distinction between DBI and DBII is lacking in the literature reviewed.

4.2.5. Celecoxib

We did not elaborate further on this explanation because the data, as explained, is preliminary.

4.3. Major Depressive Disorder

In the conclusions, we have added a paragraph dealing with the possible monitoring of the effectiveness of long-term treatments, for example by digital phenotyping.

4.4. Anxiety Disorders

We have added some considerations in this paragraph.

4.5.4. Ondansetron

-We found no data in the literature on the benefits or comparison of ondansetron with other drugs in the treatment of treatment-resistant OCD.

-We have included side effects as requested.

4.5.5. N-Acetylcysteine

Thank you for noticing, we put the N in italics.

References: We have updated the bibliography as requested by the journal.

Changes have been highlighted in yellow. In addition, a revision of the English level was carried out.

Reviewer 2 Report

Comments and Suggestions for Authors

This narrative review offers a timely and comprehensive overview of new pharmacological agents developed or approved between 2018 and 2025 for major psychiatric disorders, with a particular emphasis on treatment-resistant conditions.

Areas of Improvement

  • The manuscript would benefit from a clearer acknowledgment of the limitations inherent to the narrative review methodology, particularly regarding the absence of formal quality assessment or risk-of-bias analysis.

  • Greater caution is needed when presenting preliminary or phase II findings, in order to avoid overestimating their clinical applicability. For example:
    – Page 15, lines 182–183: "free of extrapyramidal symptoms or metabolic side effects" (referring to ulotaront);
    – Page 38, lines 1317–1318: "Psilocybin showed prolonged antidepressant effects in TRD";
    – Page 25, lines 654–655: "These findings suggest that individuals with a history of physical and/or sexual abuse may benefit from anti-inflammatory treatment" (in reference to infliximab).
    Such statements would benefit from more cautious, evidence-qualified language.

  • Sections on anxiety disorders and OCD are noticeably less developed than those on mood and psychotic disorders. Explicitly acknowledging this imbalance—and its potential reasons—would strengthen the manuscript’s transparency.

  • A more analytical appraisal of the strengths and limitations of the cited studies (e.g., sample size, trial phase, blinding, control conditions) would enhance the scientific rigor and critical depth of the review.

Author Response

Areas of Improvement

  • The manuscript would benefit from a clearer acknowledgment of the limitations inherent to the narrative review methodology, particularly regarding the absence of formal quality assessment or risk-of-bias analysis.
  • Greater caution is needed when presenting preliminary or phase II findings, in order to avoid overestimating their clinical applicability. For example:
    – Page 15, lines 182–183: "free of extrapyramidal symptoms or metabolic side effects" (referring to ulotaront);
    – Page 38, lines 1317–1318: "Psilocybin showed prolonged antidepressant effects in TRD";
    – Page 25, lines 654–655: "These findings suggest that individuals with a history of physical and/or sexual abuse may benefit from anti-inflammatory treatment" (in reference to infliximab).
    Such statements would benefit from more cautious, evidence-qualified language.
  • Sections on anxiety disorders and OCD are noticeably less developed than those on mood and psychotic disorders. Explicitly acknowledging this imbalance—and its potential reasons—would strengthen the manuscript’s transparency.
  • A more analytical appraisal of the strengths and limitations of the cited studies (e.g., sample size, trial phase, blinding, control conditions) would enhance the scientific rigor and critical depth of the review.

Thank you for your comments.

  • As requested, we have more clearly defined the selection methodology, via inclusion and exclusion criteria, of the studies included in this review.
  • Thanks to the clarification, we used more cautious language specifying when the results were preliminary or when the studies were phase I/II/III.
  • Thanks to this comment, we added the clarification in the introduction, and a hypothesis on the reasons for these differences is made explicit in the respective paragraphs.
  • The sample size, study phase, blindness, and control conditions are made explicit in the tables to make the text easier to read.

All changes have been highlighted in yellow in the text.

We revised the English language.

Reviewer 3 Report

Comments and Suggestions for Authors

Comments:

This is a comprehensive and well-structured narrative review addressing recent advancements in psychopharmacology, specifically covering new agents and formulations approved from 2018 to 2025 for major psychiatric disorders such as schizophrenia, bipolar disorder, major depressive disorder, OCD, and anxiety disorders. The manuscript is informative, clearly written, and highly relevant to clinicians and researchers working in the field of psychiatry and psychopharmacology.

The authors systematically review a broad range of medications including lumateperone, OLZ/SAM (olanzapine/samidorphan), KarXT, esketamine, dextromethorphan-bupropion, brexanolone, zuranolone, and various long-acting risperidone formulations, providing data on clinical efficacy, tolerability, and side-effect profiles. The manuscript is well-supported by recent literature and includes valuable tables summarizing key clinical trial data.

  1. The search strategy is quite exhaustive, but the inclusion and exclusion criteria are somewhat contradictory. For instance, observational studies are both included and excluded in different sections. Please clarify this in the Methods section (Lines 107–112).
  2. Since the review includes both approved and investigational agents, it would be helpful to clearly delineate between FDA/EMA-approved treatments and those still under investigation.
  3. There is inconsistency in the terminology used for certain conditions. For example, both “BD” and “bipolar disorder” are used interchangeably—choose one and maintain consistency throughout.
  4. Although the review provides detailed data on trials, a brief critical appraisal (e.g., GRADE-based commentary or at least mention of study limitations like small sample sizes or open-label design) would enhance scientific rigor.
  5. Figure 1 (PRISMA-style flow diagram) is mentioned but not properly formatted or discussed. A revised, clearly labeled figure would help readers understand the selection process.
  6. Ensure that spaces, punctuation, and line breaks are corrected. The manuscript shows several formatting inconsistencies, particularly in author information and keywords.
  7. The conclusion could benefit from a paragraph on future directions, such as the role of digital phenotyping, precision psychiatry, or pharmacogenomics.

Recommendation:

Minor Revision – The manuscript is strong and publication-worthy with minor modifications and clarifications as noted above.

Comments on the Quality of English Language
  1. Ensure that spaces, punctuation, and line breaks are corrected. The manuscript shows several formatting inconsistencies, particularly in author information and keywords.
  2. There is inconsistency in the terminology used for certain conditions. For example, both “BD” and “bipolar disorder” are used interchangeably—choose one and maintain consistency throughout.

Author Response

  1. The search strategy is quite exhaustive, but the inclusion and exclusion criteria are somewhat contradictory. For instance, observational studies are both included and excluded in different sections. Please clarify this in the Methods section (Lines 107–112).
  2. Since the review includes both approved and investigational agents, it would be helpful to clearly delineate between FDA/EMA-approved treatments and those still under investigation.
  3. There is inconsistency in the terminology used for certain conditions. For example, both “BD” and “bipolar disorder” are used interchangeably—choose one and maintain consistency throughout.
  4. Although the review provides detailed data on trials, a brief critical appraisal (e.g., GRADE-based commentary or at least mention of study limitations like small sample sizes or open-label design) would enhance scientific rigor.
  5. Figure 1 (PRISMA-style flow diagram) is mentioned but not properly formatted or discussed. A revised, clearly labeled figure would help readers understand the selection process.
  6. Ensure that spaces, punctuation, and line breaks are corrected. The manuscript shows several formatting inconsistencies, particularly in author information and keywords.
  7. The conclusion could benefit from a paragraph on future directions, such as the role of digital phenotyping, precision psychiatry, or pharmacogenomics.

Comments on the Quality of English Language

  1. Ensure that spaces, punctuation, and line breaks are corrected. The manuscript shows several formatting inconsistencies, particularly in author information and keywords.
  2. There is inconsistency in the terminology used for certain conditions. For example, both “BD” and “bipolar disorder” are used interchangeably—choose one and maintain consistency throughout.

We thank the reviewer for his comments.

1) We have clarified and removed the contradictory aspects reported in the “methods” section.

2) We have inserted a sentence in the “methods” section that clearly distinguishes which treatments have been approved by the FDA/EMA and which are still under study.

3) Thanks to the notification, we have standardized the terminology.

4) As this is a narrative review, in order not to make reading more cumbersome, we have provided this data in the tables; in addition, we have more clearly defined the selection methodology of the narrative review by making explicit the inclusion and exclusion criteria.

5) We reformatted the image according to the journal's rules. We thank you for pointing this out to us.

6) We have checked spaces, punctuation, and line breaks and corrected formatting inconsistencies.

7) Thank you for your suggestion, we have added a paragraph according to your comments. 

We have reworded some of the highlighted sentences to reduce redundancy and to improve the level of English.

Round 2

Reviewer 1 Report

Comments and Suggestions for Authors

Thank you to the authors for the revisions made. Overall, the requested changes were addressed to some extent; however, there are still some issues that need to be corrected to ensure consistency and clarity throughout the manuscript. Please find the comments below, organized by bullet points for clarity:

Line 606 – The acronym "EMA" is not introduced here for the first time; it already appears in line 137.

Line 200 – "TAAR1" is mentioned before it is properly introduced as trace amine-associated receptor 1 (line 403).

Line 180 – "dopamine (DA)" appears, but the acronym "DA" is already used earlier in line 177.

5-HT – It should be clarified that "5-HT" refers to serotonin receptors.

LAI (long-acting injectable) – The acronym first appears in Table 1, but its full form only appears later in line 191 and is then repeated in line 259. Consistency in usage is recommended.

Lines 810, 890, 904 – Replace "major depressive disorder" with "MDD" to maintain consistent use of the acronym.

Line 29 – The term treatment-resistant depression (TRD) is missing.

Line 919 – Only the acronym "TRD" should be used here, as it has already been defined earlier.

Line 963 – The inclusion of "LSD" does not seem appropriate at this point, as it is not consistently referenced in previous sections. Consider removing it for consistency.

Line 985 – The same applies to "MDMA" – it is not in line with how other substances are treated in the text.

Line 1010 – The inclusion of "DCS" is not justified, as it is not previously contextualized.

Line 985 – Correct the punctuation: remove the period between the number 3 and the comma.

Line 1142 – Replace "obsessive-compulsive disorder" with "OCD", consistent with usage elsewhere in the text.

References – The months mentioned in the references are written in Italian.

Although the article format does not require months to be included, when they are, they should not appear in Italian, nor should they be italicized.

Author Response

All changes have been done.